# Supporting Clinical COVID-19 Diagnosis with Routine Blood Tests Using Tree-Based Entropy Structured Self-Organizing Maps

Vagner Sargiani [1,*,†] , Alexandra A. De Souza [2,†], Danilo Candido De Almeida [3], Thiago S. Barcelos [2], Roberto Munoz [4] and Leandro Augusto Da Silva [1,†]

1   Laboratory of Big Data and Applied Analytical Methods—Big MAAp, Mackenzie Presbiterian University, São Paulo 01302-907, Brazil; leandroaugusto.silva@mackenzie.br
2   Laboratory of Applied Computing—LABCOM[3], Federal Institute of Education, Science and Technology of São Paulo, São Paulo 01109-010, Brazil; alexandra.souza@ifsp.edu.br (A.A.D.S.); tsbarcelos@ifsp.edu.br (T.S.B.)
3   Nephrology Division—Department of Medicine, Federal University of São Paulo, São Paulo 04021-001, Brazil; d.almeida@unifesp.br
4   Escuela de Ingeniería Informática, Universidad de Valparaíso, Valparaíso 2362905, Chile; roberto.munoz@uv.cl
*   Correspondence: vsargiani@gmail.com
†   These authors contributed equally to this work.

**Featured Application: A new algorithm that uses self-organizing maps and entropy calculus to generate a tree structure based on XAI software principles.**

**Abstract:** Data classification is an automatic or semi-automatic process that, utilizing artificial intelligence algorithms, learns the variable and class relationships of a dataset for use a posteriori in situations where the class result is unknown. For many years, work on this topic has been aimed at increasing the hit rates of algorithms. However, when the problem is restricted to applications in healthcare, besides the concern with performance, it is also necessary to design algorithms whose results are understandable by the specialists responsible for making the decisions. Among the problems in the field of medicine, a current focus is related to COVID-19: AI algorithms may contribute to early diagnosis. Among the available COVID-19 data, the blood test is a typical procedure performed when the patient seeks the hospital, and its use in the diagnosis allows reducing the need for other diagnostic tests that can impact the detection time and add to costs. In this work, we propose using self-organizing map (SOM) to discover attributes in blood test examinations that are relevant for COVID-19 diagnosis. We applied SOM and an entropy calculation in the definition of a hierarchical, semi-supervised and explainable model named TESSOM (tree-based entropy-structured self-organizing maps), in which the main feature is enhancing the investigation of groups of cases with high levels of class overlap, as far as the diagnostic outcome is concerned. Framing the TESSOM algorithm in the context of explainable artificial intelligence (XAI) makes it possible to explain the results to an expert in a simplified way. It is demonstrated in the paper that the use of the TESSOM algorithm to identify attributes of blood tests can help with the identification of COVID-19 cases. It providing a performance increase in 1.489% in multiple scenarios when analyzing 2207 cases from three hospitals in the state of São Paulo, Brazil. This work is a starting point for researchers to identify relevant attributes of blood tests for COVID-19 and to support the diagnosis of other diseases.

**Keywords:** TESSOM; self-organizing maps; entropy; XAI; visual support; COVID-19; data mining



## 1. Introduction

The use of artificial intelligence algorithms for clinical data analysis has been recently intensified to support diagnosis [1–4]. With the emergence of the COVID-19 pandemic, a great amount of data were generated, and many studies were carried out based on such

data, some of which focused on the frequency of positive cases in the city of Wuhan [5], recruitment of candidates for experimental treatments [6] and validation of effects of drugs for treatment in people diagnosed with COVID-19 [1,7,8]. To better define this data growth, the term *infodemic* is being used. According to [9], it is a combination of the terms "information" and "pandemic," serving as a reference for the excess of information (true or false) that hinders access to reliable sources of information and the definition of valid guidelines for decision making. Some studies were already carried out to help select reliable information, such as the use of epidemiological models to analyze data sources [10], where several cases of spread of misinformation were identified through the use of epidemiological models.

There are several sources contributing to the current *infodemic* related to COVID-19. Some examples come from Asiatic countries that are focused on pharmaceutical innovation [11], studies related to the macroeconomic crisis triggered by the COVID-19 pandemic [12], and also studies on the analysis of measures to help combat COVID-19 in the Russian Federation [13].

However, part of the proposed work on COVID-19 data is still based on algorithms whose focus is to improve the diagnostic prediction performance, without considering an understanding of the importance of variables and even the relationships between them in the diagnosis, making the decisions incomprehensible by specialists [14–16]. In operational terms, from a clinical point of view, it can be a problem not to give the doctor the opportunity to make the final decision [17].

Another aspect that should be highlighted, from the point of view of algorithmic limitations, is the so-called class imbalance [18]. In problems based on medical data, it is very common that more cases are related to one diagnostic outcome than to another [19]. For example, for COVID-19 cases, literature shows that there are far more data tied to positive diagnoses than negative ones. In part, this is since asymptomatic patients usually do not seek testing [5,14,20]. Therefore, a significant number of artificial intelligence algorithms have prediction results biased by class imbalance, which requires, in many of them, the application of other techniques that aim to pre-process the base in order to achieve balance, such as the prototype generation technique [21,22] or bias adjustment [23].

In addition, it should also be noted that some patients with different diagnoses may present with comorbidities, and when performing a test such as a blood test, some test factors may coincide, leading to situations in which the data indicate overlapping diagnostic outcomes—that is, the decisions are not simple. As far as this problem is concerned, artificial intelligence algorithms are particularly good generators of decision boundaries. However, in order to achieve this goal in some situations, complex mathematical operations are needed, and that makes it difficult to provide an explainable decision. This is the case with neural networks, support vector machines and other algorithms that generate a decision boundary [4,24].

The motivation for this work was two-fold. The paper [1] proposed a method based on a neural network of self-organizing maps (SOM) applied to blood counts of 599 patients diagnosed positively or negatively with COVID-19 from Albert Einstein hospital in Sao Paulo, whose results are explainable by humans, considering the paradigms defined for **XAI** (*eXplainable AI*) [25,26]. Therefore, SOM visualization methods were explored to identify the relevance of each of the 13 blood test factors (biological variables) by inspecting each neuron using linear discriminant analysis, to discover the most preponderant factors for the correct diagnosis.

The second source of motivation was the idea of attempting to exploit SOM in a similar way to AI algorithms based on decision trees, such as *random forest* and *XGBoost*. These algorithms analyze each variable of a dataset (in this case, CBC variables) in order to model the best combination of these factors based on the entropy measurement in order to maximize the accuracy for diagnosis prediction. These models may provide a clearer understanding on how a decision is made, through the interpretation of the variables in order of relevance from the top to the bottom of the tree. Disadvantageously, although the

relevance of each variable can be discovered, the algorithm in many cases ends up selecting a set of variables which does not allow a complete analysis of the variables and their relationships. Furthermore, these algorithms are sensible to class imbalance, which we knew could be a disadvantage in the context of this work. For this reason, techniques to obtain balance between classes were used to introduce artificial data, generated specifically for this purpose, that did not come from actual patients.

Therefore, this work presents a SOM network tree whose growth is controlled by entropy. Thus, the TESSOM (*tree-based entropy structured self-organizing maps*) algorithm is based on a semi-supervised training; that is, SOM training is the unsupervised portion, which is not influenced by class imbalance, and the entropy calculation for each neuron that needs labeling information is the supervised part. The outcome of this training process is the selection of neurons that should generate a new SOM network. Thus, data subsets with large overlaps, measured by entropy, form a new network in order to reorganize the data, as we seek better separation. In addition to the proposed algorithm, another contribution is the possibility of better interpreting results manually—for example, by allowing one to identify the probability of association with COVID-19 diagnosis for each range of values of each blood test factor.

To validate the algorithm and its entire interpretation structure, databases from three hospitals were used, provided by the COVID-19 Data Sharing platform (https://repositoriodatasharingfapesp.uspdigital.usp.br, accessed on 17 January 2022), an initiative of FAPESP (São Paulo State Foundation for Research Support), in cooperation with the University of São Paulo, aimed at creating a repository with data related to COVID-19 to allow research on this topic. The platform contains more than 500 types of examinations from more than 50,000 patients. After data processing, a dataset was generated containing 2453 objects distributed in 1912 positive and 541 negative cases.

The remainder of this work is organized as follows: Section 2 summarizes related works. Section 3 describes the new algorithm together with theoretical references and guidance. Section 4 contains information about the database, algorithm parameterization and the analytical process for the results. In Section 5, the results obtained in the experiments are reported, and in Section 6 a summary of the results is presented, along with perspectives for future works.

*Research Contributions*

The main goal of this work is providing a supporting tool for researchers that analyzes data while avoiding the bias introduced by the dominant class. In addition to this, the analytical process is performed in a way that is explainable to the specialist. In this way, it is possible to use TESSOM to support the identification of attributes that may contribute to COVID-19 diagnosis, and perform predictive analysis. Other goals of the model are allowing the comprehension of how the algorithm produces its outcomes and enhancing the map resolution through data segmentation. In this way, it is possible to understand the overlapping in the results of blood tests.

Hence, the main contribution of this work is the proposal of a new algorithm that allows the identification of attributes that may contribute to diagnostic explanation, enhancing the comprehension of predictive results and producing more information to allow the specialist to have more grounds for decision making. Another contribution is the segmentation of the database by similarity, disregarding the object classes. This segmentation makes it possible to identify groups of patients with different diagnostic outcomes but with similar features.

## 2. Related Works

The use of artificial intelligence algorithms in medicine is found in diverse contexts [27,28]. The works range from the detection of diseases such as cancer [29] to the analysis of clinical data [1,2,29–31]. With the epidemic caused by COVID-19, there is a large number of AI researchers participating in research on this topic, from analyzing clinical data and

understanding blood test factors caused by COVID-19 [1,32–34], to spread analysis of the disease [35] and aiding in diagnosis from blood tests [1,36–38], which was of particular interest to this work.

One study [38], which resembled the present research work in its application, analyzed AI algorithms for positive and negative diagnosis of COVID-19. In this work, the gradient boost decision tree (GBDT), random forest, logistic regression and decision tree algorithms were used, giving an average AUC ("area under the ROC curve"—away to summarize the information represented in the ROC curve) result of 83%. In addition to showing that GBDT has the best results, the authors analyzed patients segmented by age, race and sex. The main contribution was a methodology to make this type of project operational in hospitals.

In [36], blood test data for the positive and negative diagnosis of COVID-19 were used. The work used artificial neural networks, random forest, linear regression and *lasso elastic-net regularized generalized linear* (known as glmnet). The experiments were conducted with blood tests of hospitalized and symptomatic patients. As expected, the correct recognition results were higher in inpatient cases (on average 89%) than in suspected patient cases (on average 75%). As the algorithms used in the work are based on supervised learning, which are sensitive to class imbalance, and in this case, there were more objects from COVID-19 patients, the authors performed class balancing, and therefore, the results obtained already account for this type of pre-processing. In an attempt to explain the results, the authors made linear combinations of blood test factors in order to discover those most representative for diagnosis, concluding that *leukocytes*, *monocytes*, *eosinophils* and *platelets* were the most significant factors. Their diagnostic accuracy was 79%.

Using the same dataset of [32], the authors of [1] proposed an approach aimed at discovering the most representative factors of blood tests for COVID-19 diagnosis. The research was motivated by the possibility of using blood tests for diagnosis due to their fast results, low cost and presence in hospital practice. The method proposed in the analysis was based on neural networks of self-organizing maps, an unsupervised algorithm that does not require class balancing. The latter is a pre-processing procedure in data analysis that, in the context of medical cases may lead to result bias, considering that synthetic data that do in the database are introduced for the analysis of patients. This may lead to false interpretations; however, it should improve the recognition rate through algorithmic means. The authors' analysis took into account a visualization of the neurons in the map—more precisely, the importance of each factor of the blood test, weighing the findings using linear discriminant analysis (LDA), which allowed for more precision in the results' interpretation. In addition, a process of generalization analysis of the method was performed, showing in a recognition rate of 83%. The identified relevance order for blood test factors was: *leukocytes*, *basophils*, *eosinophils* and *red cell distribution width*.

The proposal of this work is a tree algorithm using self-organizing maps (SOM) to perform the classification and segmentation of data by similarity between their vector distances, according to work by [1,2,8,35], together with the use of entropy to define the pruning of the tree. Entropy has already been used by [29] for variable selection.

The next section will describe the proposed algorithm and the analysis process that can be performed with it. A comparison with other existing tree-based SOM algorithms is also presented, describing the particular aspects of this proposal.

### 3. Tree-Based Entropy Structured Self-Organizing Maps—TESSOM

*Self-organizing maps* are artificial neural networks whose architecture consists of an input layer and an output layer. The input is where an object from a dataset used for training is presented to the network. The output is usually a spatially and globally ordered two-dimensional lattice of neurons. Each neuron has a vector of weights whose dimensions are the same as the object presented in the input. The training algorithm is of the unsupervised type and has characteristics of competitiveness and cooperation. The competitiveness consists of finding the neuron that best represents the input object.

The cooperation, in turn, consists of updating the weight vector of the winning neuron to ensure that similar objects have the same neuron as the winner. The weight vector of its neighbors is also updated so that objects with less similarity have neurons adjacent to the winner. This guarantees a topological ordering of the data in the lattice. The weight vector of each of the neurons, after the model is trained, represents a virtual object that may or may not coincide with a real object, and around which the other objects of the neuron are close. This vector will be treated as a prototype. There are works such as [39,40] that deal with techniques for selecting multiple prototypes in a SOM model.

The topological ordering and also the vector quantization, given that each neuron can represent a set of similar objects, brings positive aspects, such as the analysis of data characteristics through cluster analysis and the visualization of the results using different types of graphs that allow bidimensional manipulations of the objects in a neuron (or the weight vector) for analysis and formation of groups.

Another advantage to using SOM relates to data. As presented in [39,41,42], the fact that the training is unsupervised allows ignoring class imbalance.

However, an aspect that generates uncertainty in the use of SOM has to do with defining the optimal dimensions for the map. Thus, some authors define heuristics that help with finding the best lattice dimensions [40]. However, according to [43], the size of the map to be generated directly impacts the training, so an effective approach to defining the size can be based on starting with smaller maps and then adding intermediate neurons up to the maximum size. Other authors propose structuring the SOM in a tree so that the lattice is hierarchically composed until reaching a level defined a priori.

There are two types of tree SOM structures in the literature: hierarchical and pyramidal. Hierarchical, as the name suggests, defines a tree structure with different hierarchies, where the evaluation of the values in the initial node generates a division of the objects for the secondary nodes according to their neighborhood, and so on. The pyramid organizes the dimensions of the map in a pyramidal shape, so that the top of the pyramid is the most representative node, and the other dimensions follow according to their weights.

Examples of hierarchical orderings are as follows: **BTASOM** (*binary tree TASON*) [44], where a binary tree is created, each node being a model of type **TASOM** (time adaptive self-organizing map), which is a SOM-derived algorithm that has variable learning rates and neighborhood size, and can reflect changes in the environment as per the latest input during model training [45,46]. **SOTA** (self-organizing map tree map) [47,48] has its neurons internally distributed similarly to a binary tree, with one node being considered the most important according to defined criteria, and the others organized below. **GHTSOM** (growing hierarchical tree SOM) [49] performs hierarchical model growth according to the variation of the weights in the neurons. **TTO-SOM** (tree-based topology-oriented SOM) [50] organizes neurons in a predefined tree structure, with free growth defined by the relationships between neurons and their neighborhoods. As far as pyramid arrangement is concerned, an example is **TSSOM** (tree-structured self-organizing maps) [51,52], where a multidimensional SOM has its dimensions organized in a pyramid shape using the **TSVQ** algorithm (*tree-structured vector quantization*). The topologies of some tree SOMs are exemplified in Figure 1, where (a) TSSOM, (b) SOTA, and (c) TTOSOM.

In this work, we propose *tree-based entropy structured self-organizing maps—TESSOM—* an algorithm based on a hierarchical structure based on some of the studies carried out with SOM by [1] to identify relevant variables for the correct diagnosis of patients with COVID-19 and visualizations of the SOM model [53,54], and studies with decision trees and the **k-NN** algorithm (k-nearest neighbors, classifier based on distance [22,39], where the data are grouped by proximity in relation to one among *k* objects identified by the algorithm as centroids of the groups [55]).

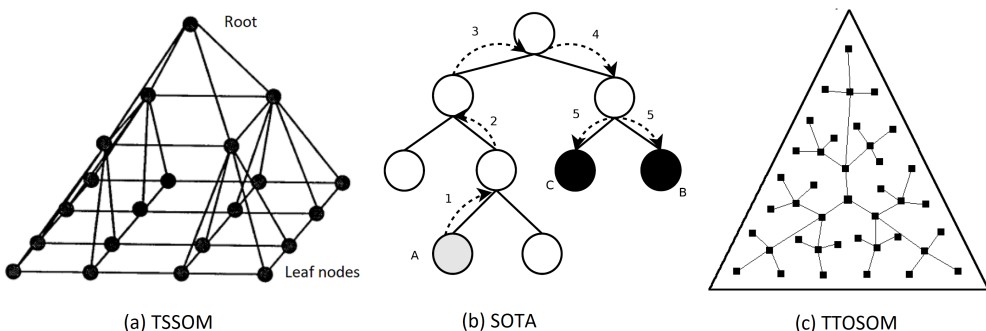

| (a) TSSOM | (b) SOTA | (c) TTOSOM |

**Figure 1.** Examples of SOM networks organized as trees [52].

The proposed algorithm, as will be detailed below, is similar to the proposals *BTASOM* and **SOTA** explained above in the sense of obtaining a tree whose segmentation of the structure is explainable and obtaining a complete model that can be analyzed at each node. However, this proposal also brings as a contribution the control of hierarchical level growth based on entropy. Thus, neurons that have entropy indicators below a threshold to be parameterized are subdivided into a new level so that objects with overlapping classes are separated.

The proposal still resembles decision tree algorithms and their variations, such as random forest. The main difference is that is it viable to identify overlapping objects from a vector perspective, whereas in the decision tree each variable is analyzed individually to better understand the effect of variables in situations that cause overlap. It is worth noting that the nature of similarity analysis in relation to the information gain calculated in the decision tree is influenced in situations of class imbalance, as is the case for the problem addressed in this work, in which the number of positive cases is greater than negative cases.

### 3.1. TESSOM Algorithm

The architecture of **TESSOM** is illustrated in Figure 2. Note that on the first level of the structure the map has a 2 × 2 structure. This is due to the fact that when considering a rectangular lattice, these are the smallest dimensions that can be configured. The structuring levels of the tree are created from each neuron, and always with the same 2 × 2 dimensions. That is, a root neuron will be subdivided in the next level, where all notes have the same 2 × 2 dimensions. In Figure 2, neurons that are within the parameters for expansion are identified in orange, and neurons that do not meet one or more parameters are identified in gray, indicating that those are "leaf" neurons; hence, they do not generate expansion. The justification for this configuration is based on algorithms for binary or quaternary trees. It is believed that in future works it will be possible to study other types of map dimensions.

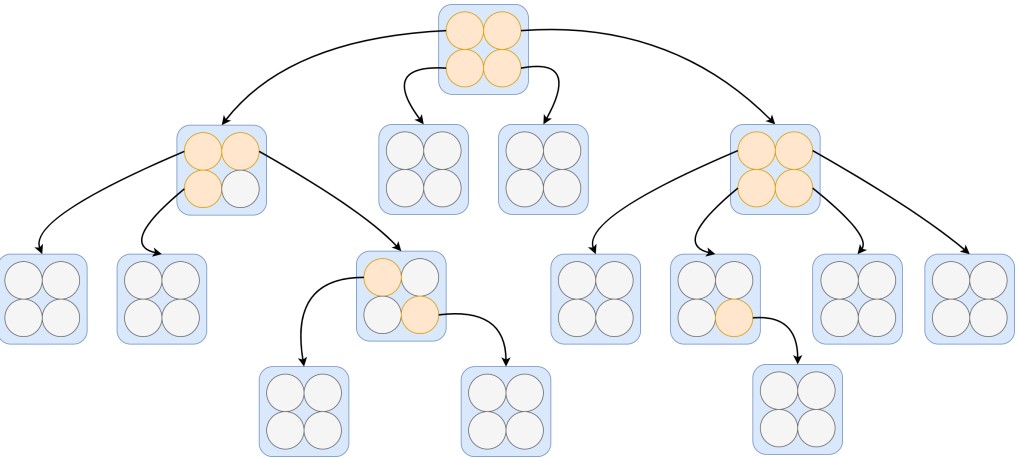

**Figure 2.** An example of a tree generated by the TESSOM model.

The fundamental process of the TESSOM algorithm is the inspection of each neuron. This means that, from a training dataset, SOM is trained using the configuration of four neurons organized in a bidimensional fashion. After training, the data subset from each neuron is analyzed using entropy to measure the level of uncertainty present in the neuron. The entropy measure for a set of two classes is performed by Equation (1), as defined by [56]:

$$H(X) = -\sum_{i=1}^{n} P(x_i) \log P(x_i) \tag{1}$$

where $P(x_i)$ is the probability calculated for the class $x_i$, and $i$ represents a class among $n$ classes.

Based on a threshold value that must be parameterized in the algorithm, which must represent the overlapping level of the classes, it is decided to continue the tree growth or stop the process as a pre-pruning of the TESSOM tree. In case the threshold value is not satisfied, the objects mapped in the neuron are separated and used to train a new $2 \times 2$ map, with the same parameterizations of the first map.

In order to ensure convergence and sufficient data for the subsequent level of the tree, a control parameter called *limObjetos* was designed, which defines a calculated value based on a percentage related to the neuron dataset under the dataset. The minimum value of the training data is parameterized and defines the early stop of growth, which is a second pre-pruning criterion.

The algorithm represented by Figure 3 works recursively, and whenever the parameters are met by a neuron, a new recursive call is made with the objects identified by this neuron as input. This code is responsible for training the model, generates views for the model information, performs some analytical processes and then performs the model entropy calculation through Algorithm 1. In this way, the model is trained until its conditions are no longer satisfied.

---

**Algorithm 1** Entrophy calculation.

---

**Require:** *Dataset* $\wedge$ *numNeurons*
**Ensure:** *EntropyDataset*
 1: *EntropyDataset* $\Leftarrow$ *newDataset*()
 2: **for** $i = 1$ to *numNeurons* **do**
 3:     *Neur* $\Leftarrow i$
 4:     *Classes* $\Leftarrow$ *distinct*(*Dataset.Neurons*[*i*].*Classes*)
 5:     *totClasses* $\Leftarrow$ *count*(*Classes*)
 6:     *rowDs* $\Leftarrow$ \{*Neur, totClasses*\}
 7:     **for** each *nomClass* in *Classes* **do**
 8:         *numObjects* $\Leftarrow$ *count*(*Dataset.Neurons*[*i*].*Classes*[*nomClass*])
 9:         *numEntropy* $\Leftarrow$ *Entropy*(*numObjects, totClasses*)
10:         *update*(*rowDS*) $\Leftarrow$ \{*nomClass, numObjects, numEntropy*\}
11:     **end for**
12:     *insert*(*EntropYDataset, rowDs*)
13: **end for**

---

As **TESSOM** is a classification algorithm, it is possible to predict the results of new objects presented to the **TESSOM** tree. To carry out this process, each new object to be evaluated is submitted to **TESSOM**. The traverses the tree, performing a prediction through the SOM model at each level, following the branches until reaching the neurons whose parameters do not allow the generation of new models, the leaves. At this point, two processes define the label prediction:

- The winning label of the neuron is identified as the label that had the highest number of occurrences at the time of training, and the test objects have their labels predicted with this result. The predicted label is confronted with the real label indicated by the object for the generation of confusion matrix (matrix that identifies the number of

objects classified correctly or incorrectly by their label), and the final result is the set of results in all leaf neurons that were subjected to test objects.

- Classifications are generated with the *kNN* algorithm, using a number of variable clusters within the range from 2 to 10, and the selection of the *kNN* cluster with the best result is done by analyzing the error curve. The cluster with the lowest value is selected. The objects used are the same ones identified by the training neuron, and the subset of new objects that, by the prediction in the **TESSOM** tree, reached the leaf neuron, are used for the test. At this point, the confusion matrix obtained by the test for this neuron will be retrieved as the values for this neuron. Again the accuracy will be based on the sum of all highlighted confusion matrices for **TESSOM**.

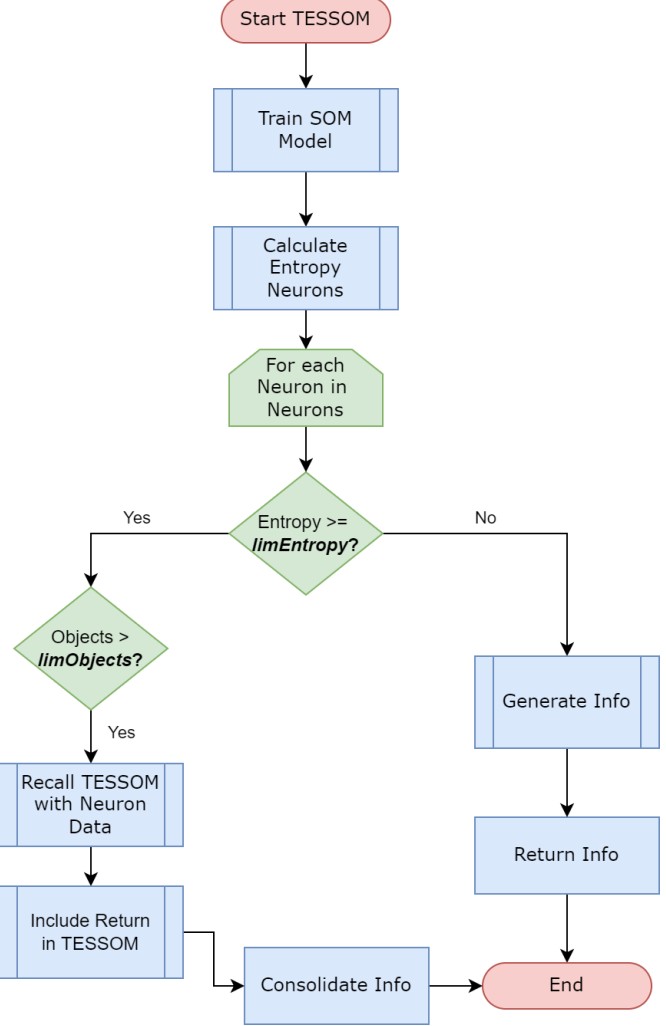

**Figure 3.** TESSOM flowchart.

This solution allows comparing accuracy results measured by various accuracy metrics in order to make adjustments to the model. As the structure for calculating accuracy is preserved, the confusion matrices can be compared at each level of the tree to identify the best results for each node of the tree.

Due to the way in which **TESSOM** processes the objects in the training set (SOM training and tree organization), the computational complexity *T* of the algorithm should be defined as a composition of complexity computations for the SOM in conjunction with its tree structure. For the SOM, as defined by [57], complexity can be defined as:

$$T = O(KMDn) \tag{2}$$

where $K$ represents the number of training epochs, $M$ is the total number of neurons in the net grid, $D$ is the number of object dimensions (number of attributes) and $n$ is the number of objects.

To complement the definition of the tree-based SOM (TESSOM), the definition presented by [58] identifies the complexity equation for decision trees, which is used in this work as the basis for calculating the asymptomatic complexity, which can be defined as:

$$T = O(n) + O(mn \log_2 n) + O(n \log_2 n) \tag{3}$$

where $n$ represents the number of objects and $m$ the number of attributes.

Assuming that $K$, $M$ and $D$ are constants, then Equation (2) can be simplified using $S = (K \times M \times D)$. Thus, the theoretical computational complexity of the TESSOM algorithm can be represented using the equation defined in (4) :

$$T = O(Sn) + O(mn \log_2(n)) + O(n \log_2(n)) + O(n) \tag{4}$$

Equation (4) summarizes the theoretical computational complexity for TESSOM, considering the batch processing used by the SOM training, the entropy calculation and the tree generation.

### 3.2. TESSOM Result Interpretation

The interpretation of the generated **TESSOM** model results can be done by analyzing the nodes (SOM models), their entropy levels and all the information collected in the process. The proposed sequence for analysis consists of the following steps:

1.  The analysis of the TESSOM tree starts at level 1, which contains the first SOM model. At this point, the database used by SOM is complete, and this model performs the first data classification process. It is possible to analyze the entropy generated and the distribution of objects by neurons, and the contributions of variables.
2.  The grouping of objects is based on vector analysis of the set of variables of the objects during training. This type of analysis makes the objects approach by close values, and for each neuron the contributions of the variables is different. When analyzing the contributions of the variables in the definition of the subset of data referring to the neuron, it is possible to perceive differences in how the data are related, and these are hidden when working with the integral base.
3.  The selection criteria for the analysis of neurons must be defined. For this example, two neurons were selected, with lower and higher entropies, to deepen the analysis. However, it is possible to expand the analysis to all neurons, as they can all have relevant information. For the analysis of the experiment of this work, the neuron with the highest number of positives per level was selected.
4.  With the neurons selected for analysis according to the criteria of the previous step, it is possible to assess which variables contributed to the clustering and which values of each variable were considered (heat map).
5.  For the neuron with the highest entropy, if it is above the limit defined in the training, it is possible to analyze the new model, this time with the data already segmented.
6.  For the neuron selected for analysis, the contribution of each variable must be evaluated, along with how the object classification process approached objects, evaluating them by their labels (remembering that they were not used in the classification process, but are available for analysis). Through the heat map, it is possible to identify the ranges of values of the variables in each of the neurons.
7.  It is possible to analyze each of the neurons whose set of objects derived a new model by selecting it and repeating the sequence from step 3, and at each new level the information contained is more and more specific. The model also provides additional information to support the analysis, such as tables containing segmented data.

Next, we exemplify how the process of analyzing the structure information generated by the algorithm **TESSOM** should be carried out using an synthetic database composed of 100 objects, containing three variables identified as: "*Attribute1*," "*Attribute2*" and a third variable containing the classes (labels) of each of the objects. The dataset was divided into three classes of 50 objects each (balanced), with overlap between the objects. The parameters used for training were: entropy of 0.5 and minimum number of neurons equal to four.

With the first SOM generated, it is possible to analyze the data overlap through the entropy table calculated for the neurons, according to Table 1 of the object mapping graph, according to Figure 4.

**Table 1.** Entropies and numbers of objects—level 1.

| Neurons | 2 | % 2 | 3 | % 3 | Total | Entropy |
|---------|-----|----------|-----|----------|-------|----------|
| 1 | 15 | 83.333% | 3 | 16.667% | 18 | 0.650022 |
| 2 | 24 | 58.537% | 17 | 41.463% | 41 | 0.97887 |
| 3 | 11 | 37.931% | 18 | 62.069% | 29 | 0.957553 |
| 4 | 0 | 0.000% | 12 | 100.000% | 12 | 0 |

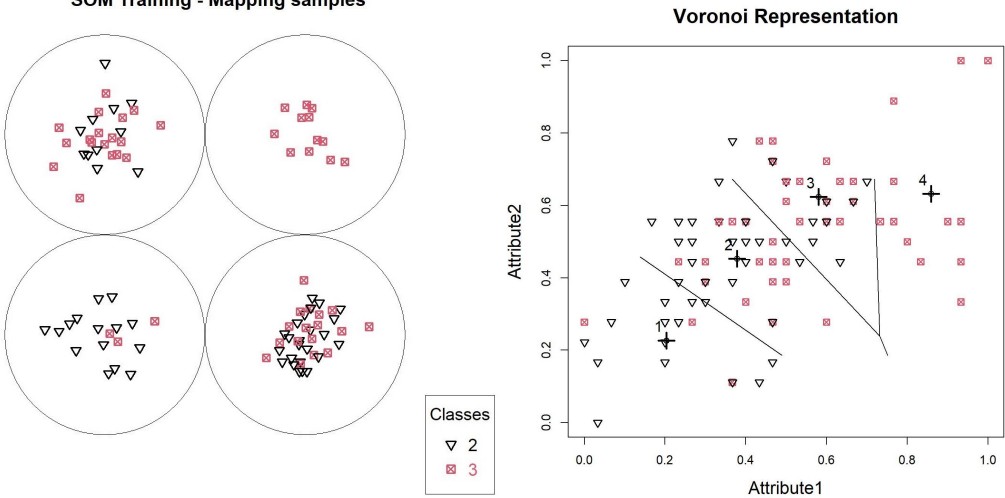

**Figure 4.** Object overlap per neuron and Voronoi representation—Synthetic database.

In this model, neurons 1, 2 and 3 have entropies above 0.5, and will generate new models. Neuron 4 contains only objects. We can validate this distribution through a Voronoi graph shown in Figure 4, where all the base points are identified, and the prototypes defined by the SOM model are identified with highlighted crosses. Lines are also printed, delimiting boundaries between objects belonging to each of the neurons.

Figure 4 shows how the definition of neurons occurred, showing that the objects were close to their prototypes, and that data overlapping did not occur for one of the neurons. To assess the participation of each variable during training, we can use the graph in Figure 5. In this figure, it is noted that in the case of neuron 4, there were equal contributions from the two variables. For the other neurons, the contribution varied, with *Attribute2* having the greatest influence.

Using the heat map shown in Figure 6, it is possible to identify which range of values of each variable contributed to each neuron. Neuron 4 was more influenced by values for *Attribute1* equal to or above 7, and for *Attribute2*, the values were equal to or above 3. This represents that objects whose values are in this range belong to Class 3 only.

## SOM Training - Weight of Variables

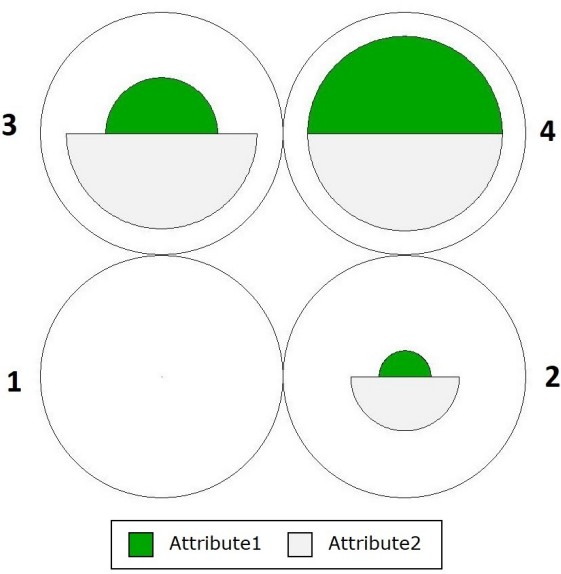

**Figure 5.** Attribute distribution—synthetic database.

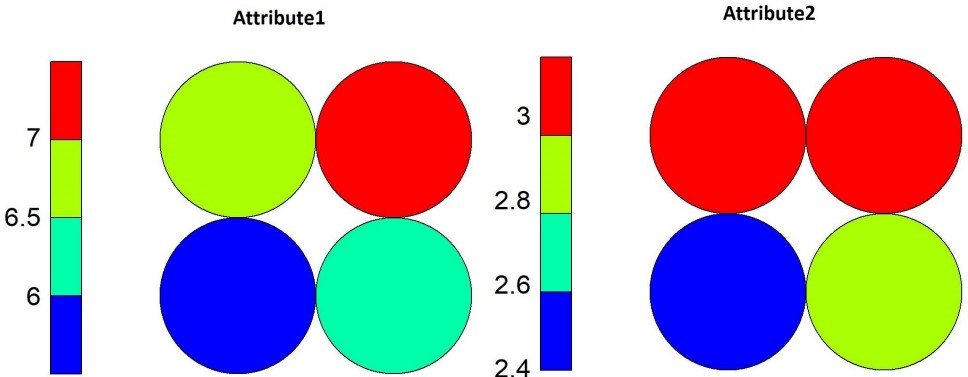

**Figure 6.** Variable heatmap—synthetic database.

The analysis proceeds to neuron 2, identified as the neuron with the greatest overlap according to the entropy calculated in Table 1. It is also possible to verify by the percentages presented that this neuron has much overlapping. The analysis was performed in the same way as for the first model.

The neuron 2 data subset allowed the generation of a new SOM model, which produced the values presented in Table 2. In this model, neuron 4 had the lowest entropy, presenting one object from the minority class and five from the majority class. However, since Equation (1) considers the relationship between the classes, the value remains high, at 0.650022. Neuron 2 showed the highest level of entropy, at 0.985228.

**Table 2.** Entropies and volumes of objects—level, 2 neuron 2.

| Neurons | 2 | % 2 | 3 | % 3 | Total | Entropy |
|---------|---|------|---|------|-------|---------|
| 1 | 7 | 63.636% | 4 | 36.364% | 11 | 0.94566 |
| 2 | 8 | 57.143% | 6 | 42.857% | 14 | 0.985228 |
| 3 | 8 | 80.000% | 2 | 20.000% | 10 | 0.721928 |
| 4 | 1 | 16.667% | 5 | 83.333% | 6 | 0.650022 |

The distribution map of the objects in the figure and the Voronoi diagram of Figure 7 demonstrate the distribution of the objects in this model. For the Voronoi diagram, the initial

scale was kept between 0 and 1 to make the comparison between the images more visual, and as SOM does, it identified the neuron objects through vector proximity, through Euclidean distance, from the neighborhood with the identified prototypes.

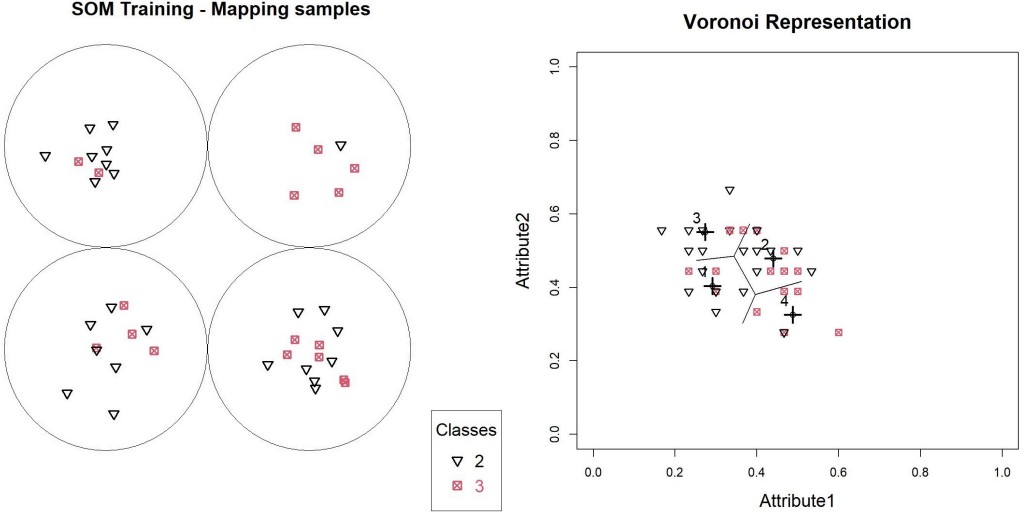

**Figure 7.** Object overlap per neuron and Voronoi representation—synthetic database, level 2, neuron 2.

When analyzing the contributions of the variables in the neurons of this model (Figure 8), we see something different from what happened with the level 1 model. The neuron that presented the lowest entropy obtained the approximation of the objects mainly via combination between high values for "*Attribute1*" and low values for "*Attribute2*," and the variable with higher entropy had similar participation by both variables.

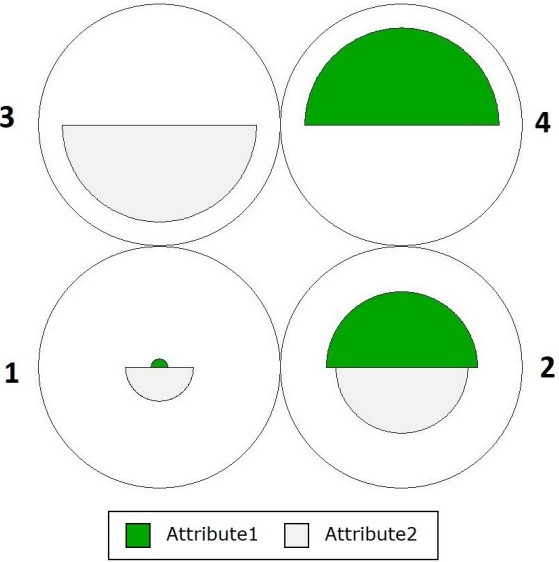

**Figure 8.** Variable distribution—synthetic database, level 2, neuron 2.

The heat maps of this model (Figure 9) indicate that, for neuron 4, values above 6.2 for "Attribute1" and less than 2.7 for "Attribute2" were the values that made the entropy of this neuron small. As for neuron 2, what contributed were values above 6.2 for the variable "*Attribute1*" and between 2.8 and 2.9 for the variable "*Attribute2*."

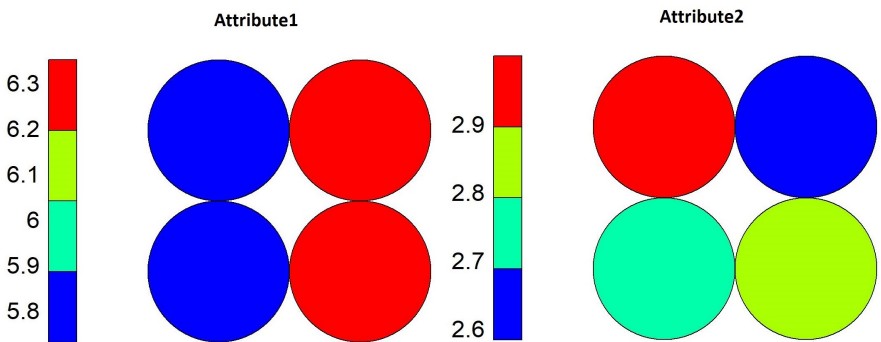

**Figure 9.** Variable heatmap—synthetic database, level 2, neuron 2.

The analysis can continue in this way in all neurons, according to the pattern or behavior that one wants to identify. In larger databases, it is possible to identify features in smaller groups that would otherwise not be identified by other classification algorithms.

## 4. Methods

The experiments in this research were performed with data from patients with positive or negative diagnoses for COVID-19 through RT-PCR examinations (reverse-transcription polymerase chain reaction). As mentioned before, we aimed to identify blood test variables that allow positive or negative diagnosis of COVID-19. The objectives were to demonstrate the effectiveness of using TESSOM in relation to other algorithms in the literature, and to help to identify prognostic scenarios based on known blood test markers for COVID-19.

### 4.1. Dataset

A public dataset (https://repositoriodatasharingfapesp.uspdigital.usp.br/handle/item/2, accessed on 17 january 2022) was used for this study. The dataset was made available on the COVID-19 Data Sharing/BR platform by two hospitals in São Paulo, Brazil: Clinics Hospital (Faculty of Medicine, University of São Paulo) on 17 February 2021 and Sírio-Libanês Hospital on 30 June 2020. More than 500 different kinds of tests from 18,424 patients allocated at the hospitals were used in the analysis, including: RT-PCR for SARSCoV-2, patient identification, patient gender, patient year of birth, patient country, federation unit and city of residence, patient test date, test description and test result. The processes of the data collection, elimination of patients' personal information and dataset generation were performed by the above-mentioned institutions.

For more accurate identification of COVID-19 cases and their severity, 2453 patient records were chosen for analysis in this study. The established selection criteria included the following rules: the first result for SARS-CoV-2 RT-PCR of the patient and the results of a blood test on the same date. These selected records contained consistent data for the following variables: hematocrit, hemoglobin, platelets, mean platelet volume, red blood cells, lymphocytes, mean corpuscular hemoglobin concentration (MCHC), leukocytes, basophils, mean corpuscular hemoglobin (MCH), eosinophils, mean corpuscular volume (MCV), monocytes and red blood cell distribution width (RDW). These variables were selected due to their frequent presence in routine automated blood test worldwide [59,60]. In the remaining records, empty data were found in one or more selected variables, and for this reason they were excluded from the pre-processing step. Of the 2453 included patients, 1912 of them tested positive and 541 negative by SARS-CoV-2 RT-PCR.

The list of blood factors that were part of this study and what they biologically represent follows:

- *Hematocrit*: Calculation of the percentage of red blood cells (also known as erythrocytes) present in the blood. A low level indicates respiratory problems, and may reflect severity in cases of COVID-19 [15,61].
- *Hemoglobin*: A substance is responsible for the transport of oxygen in the bloodstream, located in the red blood cells. A decrease in *Hb* level during the short period after

pneumonia diagnosis might be a predictor of worsening pneumonia in COVID-19 patients. In addition, anemia is a common manifestation in COVID-19 [62].

- *Red blood cells*: Blood component responsible for transporting oxygen. Changes in their shape (deformities), size or quantity can indicate anemia, problems related to hypoxia (lack of oxygenation) or even a reduction in the ability of these cells to transport oxygen [63]. RBC count is an important indicator in the evaluation of the treatment of patients with COVID-19 [17], but it is not a good indicator for detecting cases in initial examinations. In [15], time series imaging studies of the cells were performed to analyze the evolution of clinical cases related to COVID-19, and how these cells change.

- *Red blood cell distribution width* **RDW**: This is also known as the RDW coefficient of variation. It provides a measure of heterogeneity in the size of red blood cells. Originally used to identify anemia, it has also become a marker for infections and more serious diseases, such as cancer and cardiovascular and cerebrovascular diseases. In cases of COVID-19, it is not a good indicator for the disease itself, but it was identified as a marker of the severity of cases, as high RDW values are related to death, according to studies by [61,64,65].

- *Mean corpuscular hemoglobin* **MCH**: It is the average amount in each red blood cell of a protein called hemoglobin, which carries oxygen around one's body [4], and variation may be indicative of COVID-19, as reported by [66]. A low MCH value typically can indicates the presence of iron deficiency anemia. In general, patients with COVID-19 will present with slightly lower values within a standard deviation of the norm.

- *Mean corpuscular hemoglobin concentration* **MCHC**: It is similar to MCH, as MCHC is a measure of the average concentration of hemoglobin inside a single red blood cell [4]. A low MCHC shows that someone's red blood cells do not have enough hemoglobin, in part suggesting anemia. In [67], this is one of the parameters that allowed differentiating cases of COVID-19 from community-acquired pneumonia.

- *Mean corpuscular volume* **MCV**: Average size (volume) of red blood cells [4]. An increase or decrease in the average volume of red blood cells can indicate several health problems, and there are studies that have related this variation to the severity of COVID-19 [61,66]

- *Lymphocytes absolute*: Indicator of infectious processes. A low value may represent severe COVID-19, leading to early treatment or poor prognosis, according to [62];

- *Leukocytes*: White blood cells are defense cells of the immune system. It has also been identified that the COVID-19 virus can attack these cells, causing them to secrete pro-inflammatory cytokines, increasing the inflammatory status of the patient [68,69]. It is also possible to analyze markers in the genomes of white blood cells as identifiers of COVID-19 [70].

- *Basophils absolute*: Basophils are important cells in the human immune system and are normally increased in cases of allergy or prolonged inflammation. Studies indicate that they are decreased significantly in cases of infection with COVID-19, especially cases with greater severity, according to studies in [71,72].

- *Eosinophils absolute*: They are also part of the human immune system and are responsible for the body's defense against parasites and infectious agents. Like basophils, their count decreases in cases of COVID-19, according to studies in [73,74].

- *Platelets*: They are blood cells produced by the bone marrow and are responsible for the blood clotting process. This indicator should be observed carefully, as its increase does not indicate COVID-19, but may indicate complications triggered by the disease, such as thrombosis, according to [75].

- *Monocytes absolute*: According to [76], *monocytes*, and *macrophages*, are part of the body's immune system and are responsible for providing immune defense against a wide variety of microorganisms and viruses. While *macrophages* reside in body tissues, *monocytes* circulate throughout the body through the bloodstream and can be detected in blood counts. The author also pointed out that, despite being beneficial, these cells

have unfavorable effects on those infected with COVID-19, generating lesions and infections in the lung. Reference [76] reports that several studies point to a reduction in the number of *monocytes* in cases of COVID-19 infection. Studies such as [77] propose that new techniques be developed to identify the migration of these cells to the lung as a possible indicator of COVID-19 and the use of some therapies to reduce the damage caused to the organ.

- SARS_CoV2_PCR: According to [76], **RT-PCR** is an indicative test for confirmation of infection with COVID-19 because it detects the presence of the genetic material of the virus. This variable will receive the values (labels) 0 for negative cases and 1 for positive cases.

### 4.2. TESSOM Parametrization

TESSOM is based on SOM and a Shannon's entropy calculation (1) to control tree growth. The SOM parameterization for the different layers is performed considering: number of epochs equal to 300, learning rate varying between 0.0009 and 0.0001 and rectangular format with 2 dimensions of 2 neurons each ($2 \times 2 = 4$). The choice of these parameters was performed through experimentation, taking into account the size of the grid, data normalization and time for model stabilization.

The growth of the tree takes place through two parameters: level of overlap and number of objects in the neuron. In order to choose the final parameters used in this work, a sequence of executions was carried out, varying the entropy value (Table 3) and the value of the limit of objects for model generation (Table 4).

**Table 3.** TESSOM accuracy with entropy variation.

| Value | Accuracy | Level 2 | Level 3 | Level 4 | Level 5 | Level 6 |
|-------|----------|---------|---------|---------|---------|---------|
| 0.5 | 79.205% | 4 | 14 | 33 | 19 | 0 |
| 0.6 | 79.069% | 3 | 11 | 23 | 14 | 3 |
| 0.7 | 79.209% | 3 | 9 | 20 | 18 | 2 |
| 0.8 | 78.743% | 2 | 8 | 15 | 5 | 0 |
| 0.9 | 78.294% | 1 | 4 | 6 | 1 | 0 |

**Table 4.** TESSOM accuracy with variation in number of objects.

| Objects | % Objects | Accuracy | Level 2 | Level 3 | Level 4 | Level 5 |
|---------|-----------|----------|---------|---------|---------|---------|
| 25 | 0.1% | 78.743% | 2 | 8 | 15 | 5 |
| 74 | 0.3% | 78.145% | 2 | 5 | 4 | 0 |
| 123 | 0.5% | 77.982% | 2 | 4 | 0 | 0 |

As it is shown in Table 3, an increase in entropy caused a gradual reduction in the number of models generated at each level, influencing the accuracy obtained by the model. The change in the minimum number of objects, according to Table 4, directly influences the number of levels that the model will generate, pruning levels of the tree. This also directly influences the accuracy obtained.

The choice of parameters must be done with discretion, and through experimentation, always evaluating the depth of the obtained tree, the granularity of the information generated and the level of detail that is desired. Very detailed trees can generate a complex structure for further analysis, but it is rich in detail. On the contrary, very small trees may not generate enough inputs for analysis and decision making, but allow a quick understanding of how the data is segmented.

### 4.3. Performance Evaluation

The performance evaluation of the algorithm was performed using the **k-fold** cross-validation technique, as defined by [78]. The first step is to organize the objects randomly and divide them into subsets of data, where each object belongs to only one subset. The next

step is the execution of **TESSOM** training, using $k-1$ subgroups for training and 1 group for tests, performing a loop of $k$ interactions, always using different groups for tests and returning the group used previously for training. The macro flowchart for executing the processing is presented in Figure 10.

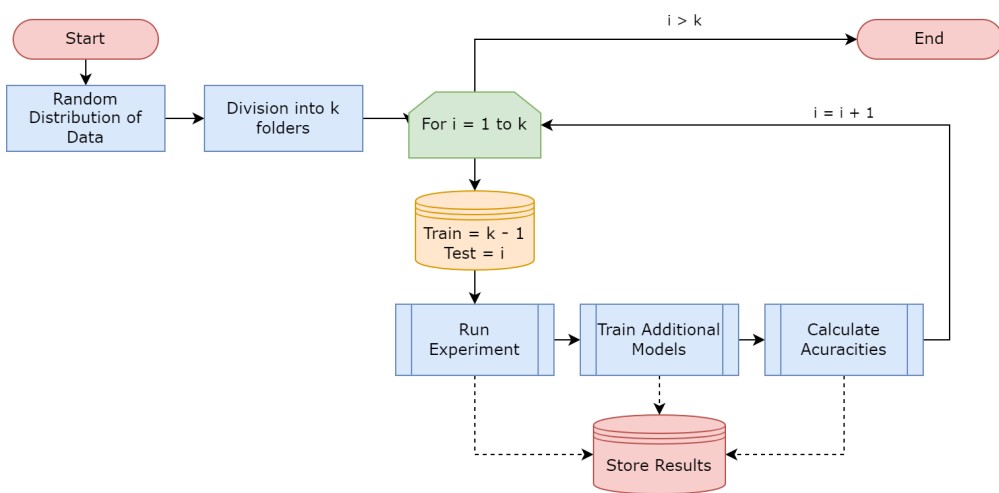

**Figure 10.** Process for experiment execution, based on cross-validation by [78].

During the training of the **TESSOM** tree, other algorithms were also used with the same subsets of objects generated by *k-folds*, as represented in Figure 10, with the collection of needed results and information. The algorithms are presented in Table 5, all with supervised training and are available for use by the library **caret**; they were also used by [79–83]. (Supervised training: during training, the labeling is considered in order to perform error corrections and increase the accuracy of the model.)

**Table 5.** Classification of comparative algorithms.

| Algorithm | Type | Based on | XAI |
|-----------|------|----------|-----|
| kNN | Classifier | Distances | No |
| Multilayer Perceptron (MLP) | Neural Network | Perceptron | No |
| Random Forest | Classifier | Decision Trees | Yes |

The reason for choosing algorithms in Table 5 for comparison is the wider use of these algorithms in research on the same topic (analysis of hemogram variables). Other algorithms based on SOM trees are related to this research, but are focused on other types of data, For example, BTASOM is used for analysis of temporal data [44] or image analysis [84].

These algorithms have different parameterizations, and the choice was made according to experiments to better adapt them to this work, or as indicated in the literature. The used parameterizations were:

- **kNN**, with $k=7$, after studying the error curve with other values, using the dataset for the experiment;
- **MLP (Multilayer Perceptron)**: 14 neurons of the hidden layer (13 variables + 1) and number of epochs = 1000 were the parameters, and there was stabilization of the model with the best results;
- **Random forest**: the number of 100 trees was defined as the best value for the algorithm according to [85,86].

For the testing phase (prediction evaluation), a test was included using the *kNN* algorithm with the complete base and using the parameterization defined above through

the **class** library. The purpose of using this library is to allow comparing the results of the testing phase of the **TESSOM** algorithm and the **caret** library.

Two types of results were evaluated: the training results and the results obtained by the **TESSOM** model. The evaluation of the algorithm training was carried out through the accuracy, calculated through data retrieved from the confusion matrix (the theoretical model is represented in Figure 11) using Equation (5), obtained during training, compared with other algorithms already used for analysis of COVID-19, which were *kNN*, *MLP* and *Random Forest* [4]. The test accuracy obtained with **TESSOM** and other models was also evaluated.

$$Accuracy = \frac{TruePositives + TrueNegatives}{AllSamples} \tag{5}$$

The evaluation of results of the **TESSOM** tree will be performed by analyzing the elements generated by the model, the generated results and conclusions obtained as described in Section 3.2. This evaluation should demonstrate how this technique allows a better interpretation of the training results and the results obtained with the classification generated by the **TESSOM** tree, according to the principles of *XAI*, in order to help with the identification of variables and/or set of variables that allow the identification of patients with COVID-19, and in some situations the severity of the cases.

|  |  | Prediction | |
|---|---|---|---|
|  |  | Positive | Negative |
| **Real** | Positive | **True Positive** | **False Negative** |
|  | Negative | **False Positive** | **True Negative** |

**Figure 11.** Interpretation of data from the confusion matrix.

Thus, the dataset with the results of blood tests of patients submitted to the RT-PCR test described in Section 4.1 was used with the **TESSOM** algorithm to perform the analysis. For this study, the following parameters were defined, selected after the analysis performed in Section 4.2:

- Entropy: 0.7;
- Minimum number of objects to generate new model: 25.

In the next section, the results of the **TESSOM** tree will be analyzed following the interpretation process described in Section 3.2 to identify relevant results in the study of COVID-19.

## 5. Results

The experiment was executed with the full dataset using the *k-fold* process, with parameter *k* set to 10. The results obtained for accuracy are presented in Table 6, which contains the results obtained for the training data, the prediction using the majority label of the neuron for the calculation and the *kNN* prediction, which is performed by generating a *kNN* classification on the objects of the neurons that were not expanded, to apply the test objects and perform the prediction.

**Table 6.** **TESSOM** training accuracy data.

|  | **Average** | **Standard Deviation** |
|---|---|---|
| Training | 79.028% | 0.4197% |
| Prediction | 76.315% | 2.9985% |
| kNN Prediction | 75.378% | 3.2374% |

With Figure 12 it is possible to evaluate the **ROC** curve for the TESSOM algorithm, which presents an **AUC** of 0.973 for training (green line) and 0.749 for model tests (prediction—red

line). Given the curves obtained bu *MLP*, *kNN* and *Random Forest*, presented in Figure 13, the gain obtained through the use of TESSOM can be identified.

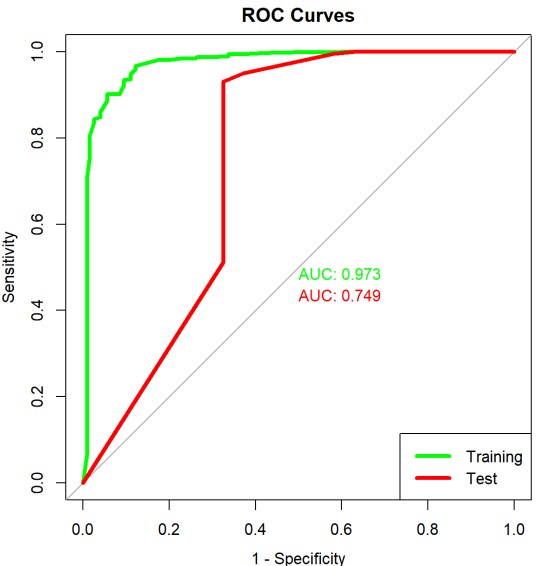

**Figure 12. TESSOM** ROC curve for training and testing.

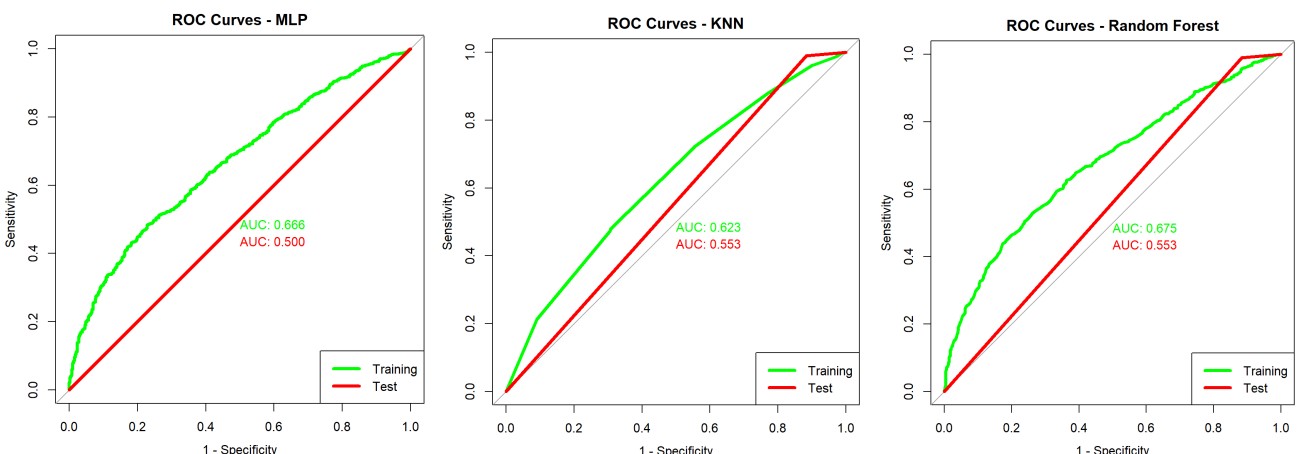

**Figure 13.** ROC curves for *MLP*, *kNN* and *Random Forest* models.

The execution times of each algorithm in seconds are shown in Table 7. It is important to note that for prediction, TESSOM presents higher times because it involves two methods, and one of the methods internally has *kNN* training that uses the objects in the neuron to perform the prediction.

**Table 7.** Algorithms execution times (seconds).

| Algorithm | Train | Prediction |
|-----------|-------|-----------|
| TESSOM | 26.907153 | 6.354581 |
| kNN | 1.078941 | 0.022347 |
| Multilayer Perceptron (MLP) | 8.653341 | 0.008462 |
| Random Forest | 36.464645 | 0.022636 |

The accuracy values obtained for all models trained and used for validation are identified in two tables: in Table 8 are the results obtained by using the training data, and in Table 9 the results obtained using the test set are presented.

**Table 8.** Accuracy comparison—models trained with the complete dataset.

| Model | Average | Standard Deviation |
| --- | --- | --- |
| TESSOM | 79.028% | 0.443% |
| kNN | 76.745% | 0.605% |
| MLP | 77.968% | 0.310% |
| Random Forest | 77.905% | 0.378% |

**Table 9.** Accuracy comparison—models trained with test set.

| Model | Average | Standard Deviation |
| --- | --- | --- |
| TESSOM | 76.315% | 3.161% |
| TESSOM-NN | 75.378% | 3.415% |
| MLP | 77.986% | 2.738% |
| Random Forest | 77.621% | 2.867% |

By evaluating the obtained results, one can observe that the **TESSOM** model presented better accuracy levels of $\approx 2\%$ in comparison to the other algorithms used in the training phase. In addition, its prediction accuracy is equivalent to that of other models, when considering the standard deviation.

However, the objective of these experiments was to analyze the performance of the proposed algorithm in comparison to other alternatives, emphasizing that the greatest contribution to be presented is in the possibility of interpreting the **TESSOM** results. Next, the results of the **TESSOM** tree are analyzed, following the interpretation process described in Section 3.2 to identify useful results for the COVID-19 study.

**TESSOM**, as the name implies, generates a tree structure that can be interpreted in a top-down manner. The first SOM model generated is considered as the root node (or level 1) of the model. Models derived from the subsets of data identified by their neurons are level 2, and so on. In Figure 14, all results obtained at the different levels of the **TESSOM** tree, up to the third level, are presented.

To demonstrate the use of **TESSOM** as an aid to identify variables that contribute to the identification of positive cases of COVID-19, the analysis focuses on neurons with percentages of positive cases higher than 80%, and whose volume of associated objects is higher than 50. These values are defined because a low number of objects may not statistically represent relevant information, considering that the total set has 2453 objects.

When performing the analysis of the results shown in Figure 14, it could be seen that at level 1, neuron 4 had an entropy of $\approx 0.62$ and a percentage of positives of $\approx 84.7\%$, according to Table 10. Therefore, the analysis started at this neuron. It had a total of 748 objects, 633 labeled as positive and 115 as negative. Neuron 4 satisfied the algorithm's criteria, and therefore did not need to be decomposed into another map.

For the purpose of illustrating the interpretation of results, an analysis of how information can be extracted from this neuron is presented. It is possible to evaluate how the organization process was performed by the neurons in level 1. It is convenient to use graphical representations to aid this analysis, such as the distribution of objects and the weights of variables in training, which are available in Figure 15. Through these graphs, we can evaluate how object grouping was performed, and which variables had the highest weights during training. Another important graph that should be considered during the analysis process is the heat map, as shown in Figure 16, which demonstrates the individual weights of variables in the SOM model.

| Level 1 | 0 | % 0 | 1 | % 1 | Total | Entropy | Level 2 | 0 | % 0 | 1 | % 1 | Total | Entropy | Level 3 | 0 | % 0 | 1 | % 1 | Total | Entropy |
|---|---|---|---|---|---|---|---|---|---|---|---|---|---|---|---|---|---|---|---|---|
| 1 | 41 | 33.065% | 83 | 66.935% | 124 | 0.9155842 | 1 | 35 | 33.019% | 71 | 66.981% | 106 | 0.9151190 | 1 | 0 | 0.000% | 7 | 100.000% | 7 | 0.0000000 |
| | | | | | | | | | | | | | | 2 | 18 | 46.154% | 21 | 53.846% | 39 | 0.9957275 |
| | | | | | | | | | | | | | | 3 | 7 | 25.000% | 21 | 75.000% | 28 | 0.8112781 |
| | | | | | | | | | | | | | | 4 | 10 | 31.250% | 22 | 68.750% | 32 | 0.8960382 |
| | | | | | | | 2 | 1 | 100.000% | 0 | 0.000% | 1 | 0.0000000 | - | - | - | - | - | - | - |
| | | | | | | | 3 | 0 | 0.000% | 2 | 100.000% | 2 | 0.0000000 | - | - | - | - | - | - | - |
| | | | | | | | 4 | 5 | 33.333% | 10 | 66.667% | 15 | 0.9182958 | - | - | - | - | - | - | - |
| 2 | 116 | 28.713% | 288 | 71.287% | 404 | 0.8649833 | 1 | 23 | 33.824% | 45 | 66.176% | 68 | 0.9231200 | 1 | 6 | 25.000% | 18 | 75.000% | 24 | 0.8112781 |
| | | | | | | | | | | | | | | 2 | 2 | 66.667% | 1 | 33.333% | 3 | 0.9182958 |
| | | | | | | | | | | | | | | 3 | 12 | 40.000% | 18 | 60.000% | 30 | 0.9709506 |
| | | | | | | | | | | | | | | 4 | 3 | 27.273% | 8 | 72.727% | 11 | 0.8453509 |
| | | | | | | | 2 | 19 | 25.676% | 55 | 74.324% | 74 | 0.8218127 | 1 | 1 | 50.000% | 1 | 50.000% | 2 | 1.0000000 |
| | | | | | | | | | | | | | | 2 | 4 | 44.444% | 5 | 55.556% | 9 | 0.9910761 |
| | | | | | | | | | | | | | | 3 | 4 | 33.333% | 8 | 66.667% | 12 | 0.9182958 |
| | | | | | | | | | | | | | | 4 | 10 | 19.608% | 41 | 80.392% | 51 | 0.7140153 |
| | | | | | | | 3 | 37 | 23.718% | 119 | 76.282% | 156 | 0.7903183 | 1 | 0 | 0.000% | 3 | 100.000% | 3 | 0.0000000 |
| | | | | | | | | | | | | | | 2 | 7 | 20.588% | 27 | 79.412% | 34 | 0.7335379 |
| | | | | | | | | | | | | | | 3 | 18 | 25.352% | 53 | 74.648% | 71 | 0.8168114 |
| | | | | | | | | | | | | | | 4 | 12 | 25.000% | 36 | 75.000% | 48 | 0.8112781 |
| | | | | | | | 4 | 37 | 34.906% | 69 | 65.094% | 106 | 0.9332227 | 1 | 14 | 28.571% | 35 | 71.429% | 49 | 0.8631206 |
| | | | | | | | | | | | | | | 2 | 11 | 28.205% | 28 | 71.795% | 39 | 0.8582308 |
| | | | | | | | | | | | | | | 3 | 7 | 63.636% | 4 | 36.364% | 11 | 0.9456603 |
| | | | | | | | | | | | | | | 4 | 5 | 71.429% | 2 | 28.571% | 7 | 0.8631206 |
| 3 | 226 | 24.275% | 705 | 75.725% | 931 | 0.7995832 | 1 | 36 | 27.273% | 96 | 72.727% | 132 | 0.8453509 | 1 | 1 | 16.667% | 5 | 83.333% | 6 | 0.6500224 |
| | | | | | | | | | | | | | | 2 | 19 | 27.143% | 51 | 72.857% | 70 | 0.8435071 |
| | | | | | | | | | | | | | | 3 | 3 | 30.000% | 7 | 70.000% | 10 | 0.8812909 |
| | | | | | | | | | | | | | | 4 | 13 | 28.261% | 33 | 71.739% | 46 | 0.8589810 |
| | | | | | | | 2 | 74 | 19.577% | 304 | 80.423% | 378 | 0.7133813 | 1 | 11 | 13.415% | 71 | 86.585% | 82 | 0.5687009 |
| | | | | | | | | | | | | | | 2 | 14 | 12.963% | 94 | 87.037% | 108 | 0.5564216 |
| | | | | | | | | | | | | | | 3 | 29 | 26.364% | 81 | 73.636% | 110 | 0.8321843 |
| | | | | | | | | | | | | | | 4 | 20 | 25.641% | 58 | 74.359% | 78 | 0.8212809 |
| | | | | | | | 3 | 52 | 34.667% | 98 | 65.333% | 150 | 0.9310558 | 1 | 3 | 37.500% | 5 | 62.500% | 8 | 0.9544340 |
| | | | | | | | | | | | | | | 2 | 16 | 32.653% | 33 | 67.347% | 49 | 0.9113424 |
| | | | | | | | | | | | | | | 3 | 28 | 34.146% | 54 | 65.854% | 82 | 0.9262122 |
| | | | | | | | | | | | | | | 4 | 5 | 45.455% | 6 | 54.545% | 11 | 0.9940302 |
| | | | | | | | 4 | 64 | 23.616% | 207 | 76.384% | 271 | 0.7886000 | 1 | 27 | 23.894% | 86 | 76.106% | 113 | 0.7932698 |
| | | | | | | | | | | | | | | 2 | 27 | 22.131% | 95 | 77.869% | 122 | 0.7625550 |
| | | | | | | | | | | | | | | 3 | 9 | 37.500% | 15 | 62.500% | 24 | 0.9544340 |
| | | | | | | | | | | | | | | 4 | 1 | 8.333% | 11 | 91.667% | 12 | 0.4138169 |
| 4 | 115 | 15.374% | 633 | 84.626% | 748 | 0.6191292 | - | - | - | - | - | - | - | - | - | - | - | - | - | - |

**Figure 14.** Training information **TESSOM** from levels 1, 2 and 3.

**Table 10.** Entropy and volume of objects per neuron, level 1 **TESSOM**.

| Neuron | 0 | % 0 | 1 | % 1 | Total | Entropy |
|---|---|---|---|---|---|---|
| 1 | 41 | 33.065% | 83 | 66.935% | 124 | 0.915584 |
| 2 | 116 | 28.713% | 288 | 71.287% | 404 | 0.864983 |
| 3 | 226 | 24.275% | 705 | 75.725% | 931 | 0.799583 |
| 4 | 115 | 15.374% | 633 | 84.626% | 748 | 0.619129 |

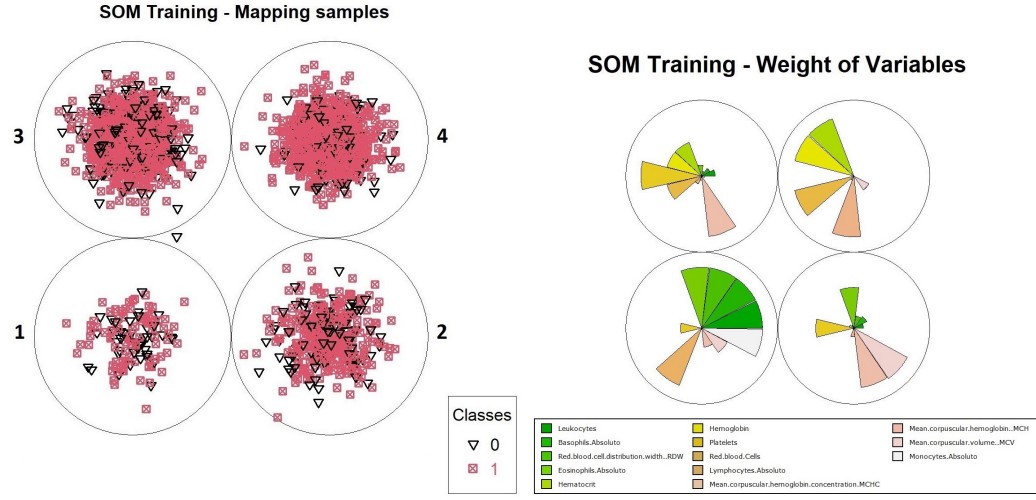

**Figure 15.** Distribution objects and weights per neuron, level 1 **TESSOM**.

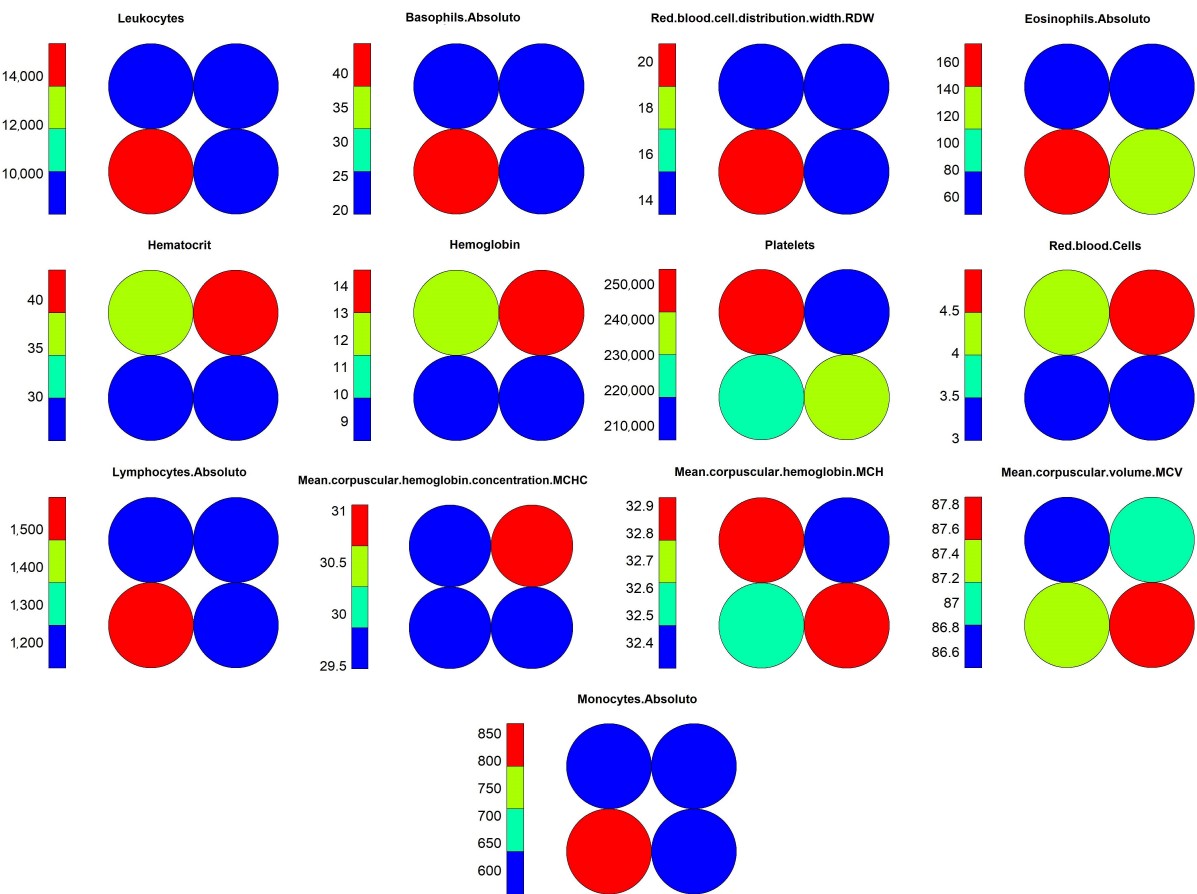

**Figure 16.** Heatmap of the variables per neuron level 1 **TESSOM**.

The interpretation of results for the level 1 model continues with visual support from Figures 15 and 16, where one can identify the occurrence of a common pattern in some variables in neuron 4 and its neighbors, and that differentiates them from neuron 1, in which positive and negative cases were overlapping within the model. The variables which composed the pattern were: "Leukocytes'," "Basophils.Absolute," "RDW", "Lymphocytes.Absolute" and "Monocytes.Absolute". This pattern can be found on level 1 more noticeably, but it is much less evident on the other levels because the division of the dataset into smaller subsets by an association of the objects with the neurons caused other variables to become more relevant, hiding the pattern. For this reason, the pattern analysis was performed only at this level along with the comparison with the most overlapping neuron, which was neuron 1.

Moving on with the analysis, it is possible to identify which variables helped in differentiating the objects in neuron 4 from the others. These variables were: "Hematocrit", "Hemoglobin", "Platelets" "Red.blood.Cells", "MCHC" and "MCH". Identifying this set of variables may be important to support decision making during the diagnosis of positive cases, considering the other variables that showed a similar pattern of weights.

To validate the obtained result, a comparison was performed with the 95% confidence intervals for the dataset. The results are available in Figure 17, where the confidence intervals for positive (label 1, in red) and negative (label 0, in black) for the identified variables are represented. From these results, one can compare the ranges of values obtained for each of these variables, based on the minimum and maximum values, between the two neurons identified in the **TESSOM**.

From this analysis, we highlight neuron 4 for its high rate of positivity, and for comparison, basis neuron 1, which had the highest value of entropy, at ≈0.91, indicating a high degree of overlapping (consequently, lower number of positives). These results are

available in Table 11, where it is possible to see the differences between the value ranges for each variable. Another relevant aspect is the positivity rate present in neuron 1. Note that it was the lowest rate of all neurons represented in the map, as only $\approx 66.9\%$ of objects were labeled as positive. It is important to reinforce that both neurons had negative and positive values. However, in neuron 4 one can identify a higher percentage of positives; hence, it may be possible to effectively evaluate behaviors in variables that help identify positivity. Additionally, this neuron had more associated objects (748) than neuron 1 (124).

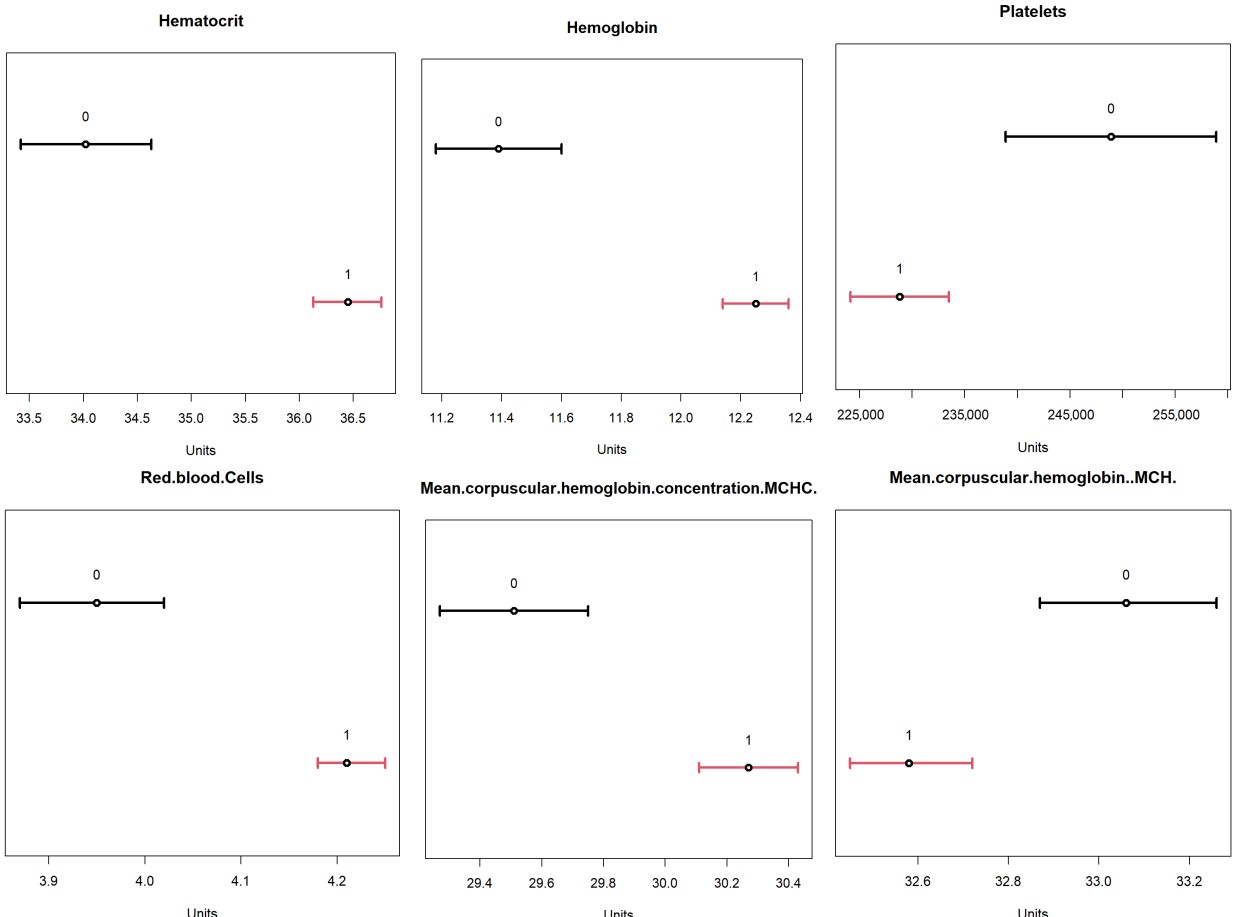

**Figure 17.** 95% confidence intervals of the variables—level 1—**TESSOM**.

**Table 11.** Minimum and maximum values of variables found in neurons 1 and 4 of level 1 **TESSOM**.

| | Neuron 1 | | Neuron 4 | |
| --- | --- | --- | --- | --- |
| **Variable** | **Minimum** | **Maximum** | **Minimum** | **Maximum** |
| Hematocrit | 13.5 | 38.6 | 36.9 | 68.6 |
| Hemoglobin | 5.1 | 14.6 | 12.3 | 19.8 |
| Platelets | 12,000 | 825,000 | 24,000 | 632,000 |
| Red.blood.Cells | 1.32 | 4.79 | 3.83 | 7.93 |
| MCHC | 16.9 | 110.6 | 21 | 37.9 |
| MCH | 20.7 | 108.1 | 21.2 | 39.7 |

The comparison between the variables of neurons 1 and 4 of level 1 allowed us to find that the previously identified variables in fact influenced the differentiation of diagnoses in the neurons, providing for neuron 4 a higher percentage of positives.

The comparison of the value ranges obtained from neuron 4 (Table 11) with the confidence ranges shown in Figure 17 for the same neuron was performed to assess whether

the value ranges of the identified variables present the same profile shown by the confidence interval. For example, "Hematocrit" had a confidence interval between 36 and 37 for positives and one between 33.5 and 34.6 for negatives; and neuron 4 had a range of values between 36.9 and 68.6. It is important to remember that, as neuron 4 dealt with a subset of the data, not all objects used in the calculation of confidence intervals were available, so the values found for the neuron may vary. What is being analyzed here is the tendency that the variable found has to present a differentiating factor in the neuron that has a relation to the total dataset represented in the confidence intervals.

The analysis of the neurons that can allow the search for variables to identify positive cases should proceed to level 2 of the **TESSOM**. By analyzing Figure 14, one can identify the generation of three models, derived from neurons 1, 2 and 3 of level 1. When analyzing the percentages of positives and negatives of the neurons for the three models, one can see that only neuron 2, of the models generated from the subset of data obtained from neuron 3 (level 1), met the established requirements (positive objects with $\geq 80\%$ and the number of objects $\geq 50$). For this neuron, the values found were $\approx 0.71$ entropy, a value that does not meet the algorithm's criteria and caused it to be decomposed into another map. The percentage of positive results was $\approx 80.5\%$, there being 304 positive objects in the subset total, that is, of 378 objects. The graphs of object distribution and variable weights are available in Figure 18.

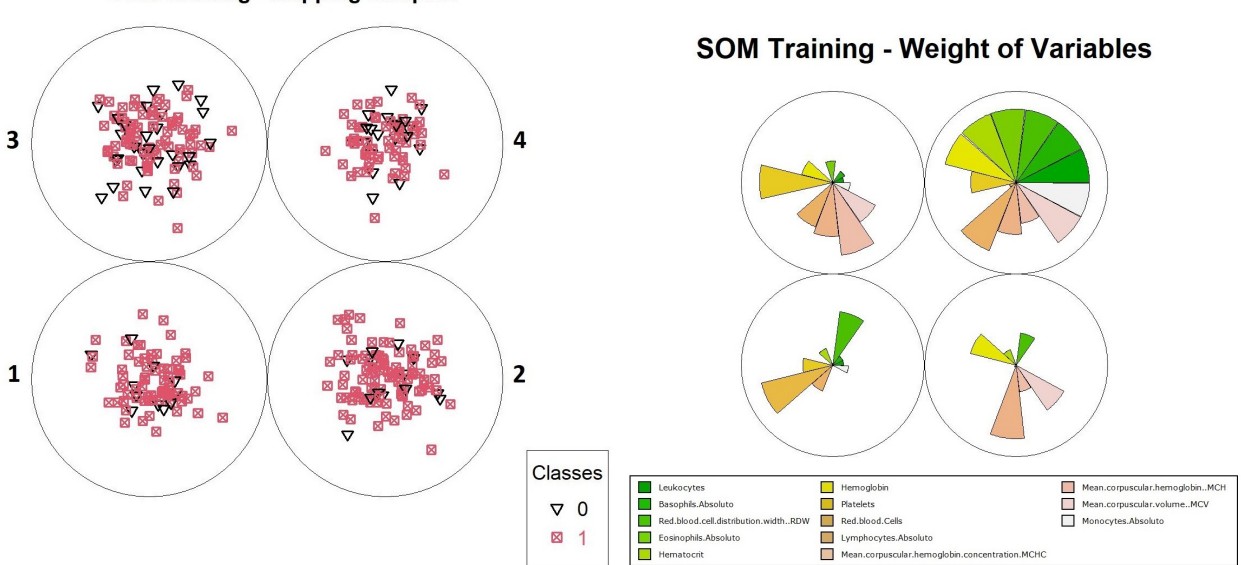

**Figure 18.** Distribution of objects and weights per neuron—level 2 **TESSOM**.

When performing a comparative analysis of the weight distribution in Figure 18 to the heat map for the same TESSOM model of level 2 (Figure 19), it is not possible to identify the same patterns of variables identified at level 1 for the reasons explained earlier, but it is possible to see that the variables that contributed to the differentiation of neuron 2 were "Platelets", "Lymphocytes.Absolute" and "MCHC", because of their weights.

When analyzing the variables found in the differentiation of this neuron with the confidence intervals for them (Figure 20) using the values available in Table 12, it can be seen that the variables "Platelets," "Lymphocytes.Absolute" and "MCHC" had the expected trend according to the confidence intervals for positives of the complete dataset. This comparison should be seen only as an evaluation of the behavior of the variables that differentiated this neuron from the database.

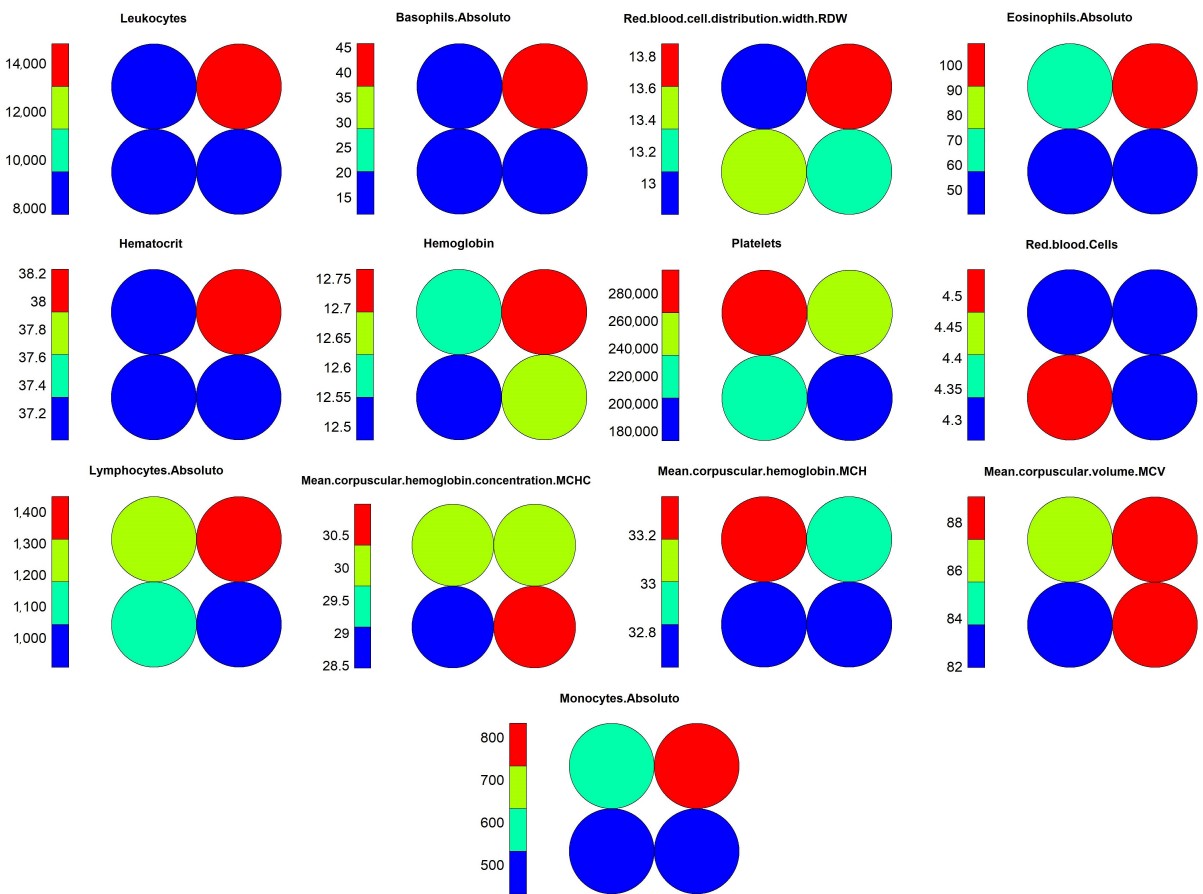

**Figure 19.** Heatmap of the variables per neuron level 2 **TESSOM**.

**Table 12.** Minima and maxima of the variables found from neuron 2, level 2 **TESSOM**.

| Variable | Minimum | Maximum |
|---|---|---|
| Platelets | 68,000 | 379,000 |
| Lymphocytes.Absoluto | 180 | 14350 |
| MCHC | 25.1 | 36.1 |

To proceed with the analysis for level 3, it is necessary to identify which neurons match the defined criteria (positive objects with percentage ≥80% and the number of objects ≥50). For these conditions, three neurons were identified, which are described below:

1. Neuron 4, with a subset of objects obtained from level 1 → neuron 2 and level 2 → neuron 2. Percentage of positive objects of ≈80.392% and calculated entropy of 0.71. Total of 51 associated objects, 41 of which are positive. Given its entropy, this neuron will be decomposed into a new model.
2. Neuron 1, with subset of objects obtained from level 1 → neuron 3 and level 2 → neuron 2. Percentage of positive objects of ≈86.59% and calculated entropy of 0.568. Total of 82 objects, and 71 positives. Will not be decomposed into a new model.
3. Neuron 2, with a subset of objects obtained from level 1 → neuron 3 and level 2 → neuron 2. Percentage of positive objects of ≈87.037% and calculated entropy of 0.556. Total of 108 objects, 94 positives. Was not decomposed into a new model.

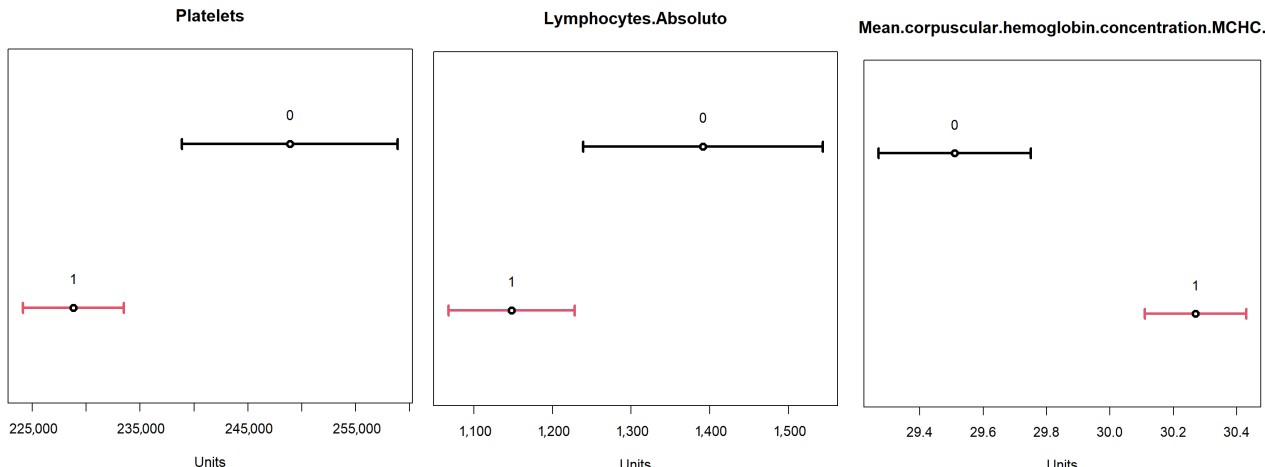

**Figure 20.** 95% confidence intervals of the variables identified at level 2—**TESSOM**.

After the identification of viable neurons for analysis, they were pre-evaluated to select which ones could be analyzed. Two neurons were derived from the neuron previously analyzed on level 2 (neuron 2 of the model generated from neuron 3 of level 1). Since the source neuron has already been analyzed because it has a high volume of positives, these neurons will not be analyzed, and only neuron 4 will be analyzed, which is the first item in the list above, where its features have been described. To continue with the analytical process, one should use the graph of weights of the variables of the model available in Figure 21 and compare it with the heat map obtained for this same model available in Figure 22. Since at level 3 the objects had already been segregated at level 1 and level 2 previously, on this level the values of the variables for the objects were closer, so identifying the differentiating variables could involve selecting those that differentiated two neurons at the same time, and the segregation was obtained by combining the different variables in different neurons. Thus, it can be identified that the variables that contributed to differentiation were "Basophils.Absolute", "Platelets", "Red.blood.Cells", "Lymphocytes.Absolute", "MCH" and "MCV".

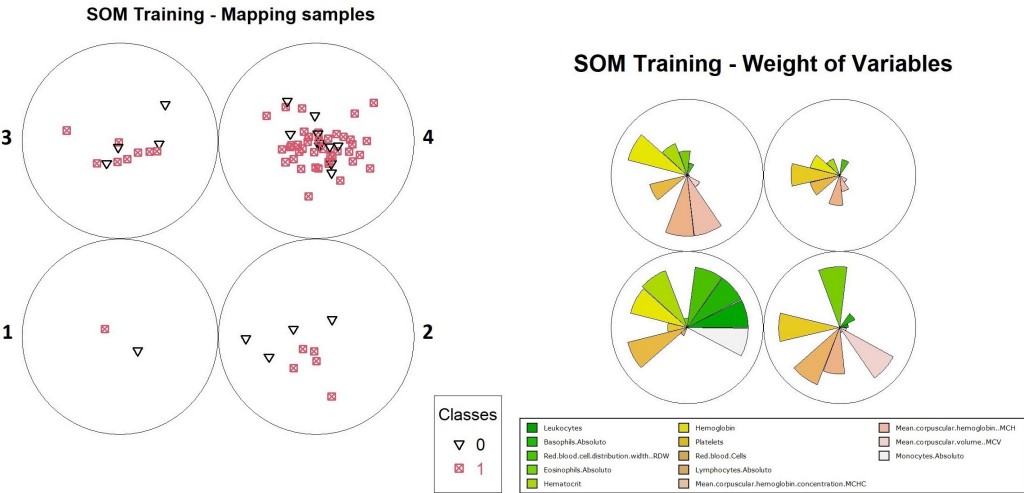

**Figure 21.** Distribution of objects and weights per neuron—level 3 **TESSOM**.

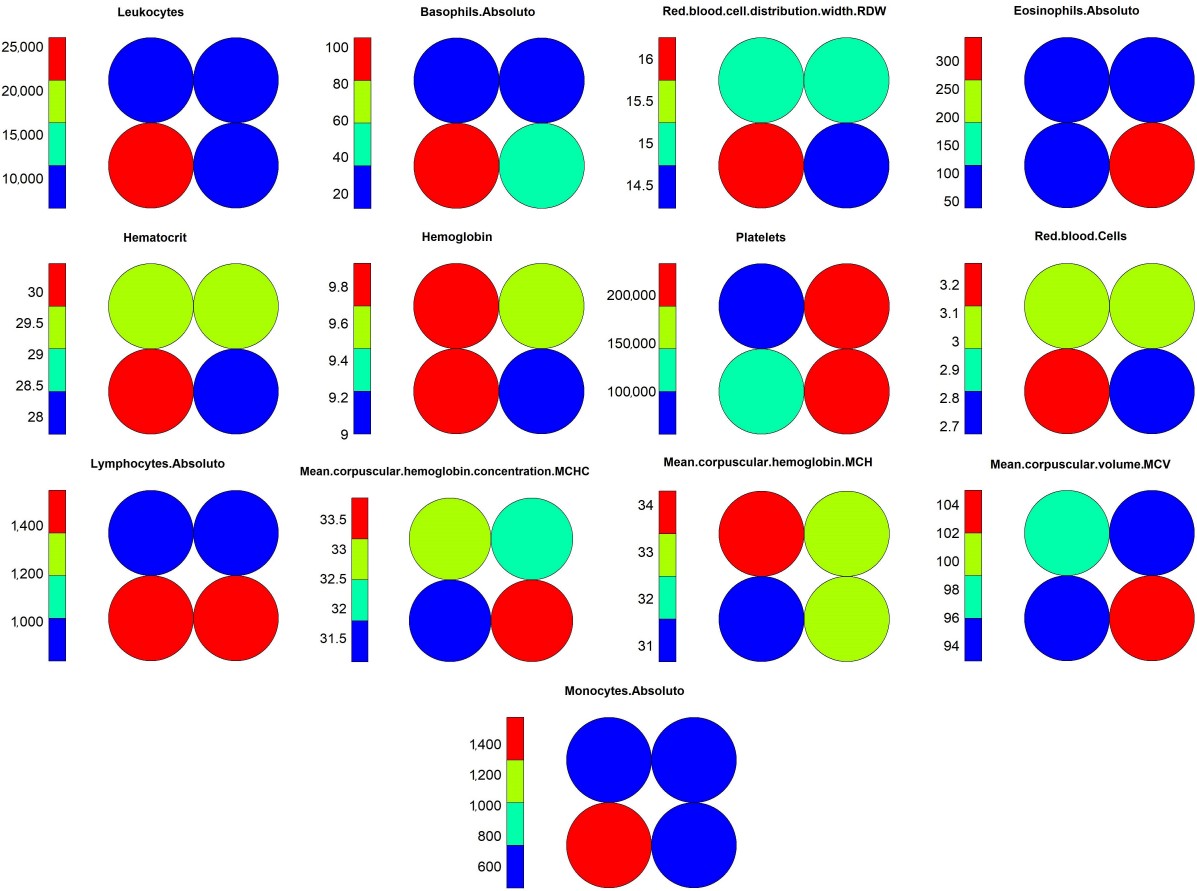

**Figure 22.** Heatmap of the variables per neuron in level 3—**TESSOM**.

When comparing the minimum and maximum values for variables in for neuron 4, presented in Table 13, with the confidence intervals of the dataset for these variables, as shown in Figure 23, we see that the values differed because the subset of this neuron had 51 associated objects out of a total of 2453 objects in the dataset, being a refinement of the obtained data. Eventually, it is possible to verify that the trend of the values of the identified variables is related to the confidence intervals discriminated for them.

**Table 13.** Minima and maxima of the variables found for Neuron 4, level 3 **TESSOM**.

| Variável | Minimum | Maximum |
|---|---|---|
| Basophils.Absoluto | 0 | 60 |
| Platelets | 84,000 | 343,000 |
| Red.blood.Cells | 2.44 | 3.52 |
| Lymphocytes.Absoluto | 80 | 2,000 |
| MCH | 29 | 36.5 |
| MCV | 89.4 | 108.4 |

After the analysis of the neurons of interest was performed, Table 14 was created, containing all the variables used in the study, showing in which neurons each variable contributed to the differentiation of positive cases for COVID-19. In this table, it is possible to see that several variables can significantly contributed to the identification of positive cases.

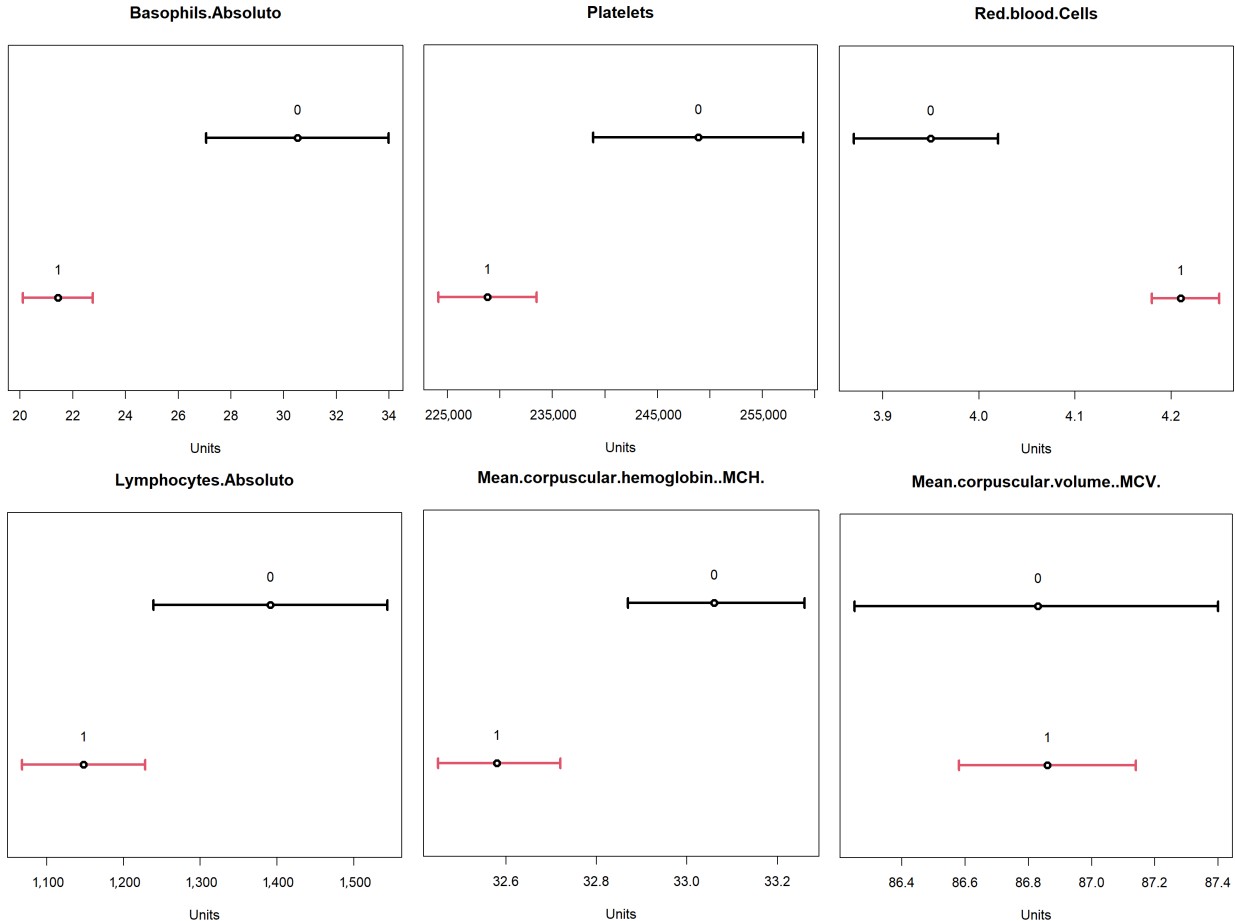

**Figure 23.** 95% confidence intervals of the variables identified at level 3—**TESSOM**.

Specifically, it is important to consider that some CBC variables found in this work appeared coincident in two of the three levels of the analyzed decomposition, and may be more representative for decision making in the clinical diagnosis of COVID-19. Among these, we can highlight platelets (3×), red blood cells (2×), lymphocytes (2×), MCHC (2×) and MCH (2×). Interestingly these variables, as already mentioned in Section 4.1, have associations with the pathophysiology of COVID-19 and have already been correlated with platelet hyperactivation and thrombosis; anemia and iron metabolism deficiency; and lymphocyte loss by viral attack (lymphopenia), pneumonia and inflammation [87–89]. It is also possible to emphasize that **TESSOM**, by using all variables at the same time and entropy to generate the tree structure, allows the identification of which attributes are most relevant for the sought diagnosis. This contrasts with the relevance of the **random forest** variables, presented in Table 15. Besides the fact that contributing variables for differentiation were different in each algorithm, it is necessary to consider that for **TESSOM** an expected result was sought among positive cases, and for **random forest** this selection is not possible.

Thus, it is possible to use TESSOM as an important auxiliary tool for identifying variables that can serve as auxiliary predictors to facilitate decision making in the identification of positive cases of COVID-19. As an additional advantage, using TESSOM it is possible to identify ranges of values for the variables which held classification power, in each subset of data created by each neuron. The identification of this range of values acts as additional support to assist decision making or even to identify the progression of the disease. It is important to emphasize that this is a data analysis work and that any conclusions regarding the identification of positive cases should be supported with the appropriate clinical studies performed by health professionals and do not replace the gold standard diagnostic test, such as RT-PCR.

**Table 14.** Summary of the variables found in the neuron analysis—**TESSOM**.

| Variable | Neuron 4 | Neuron 3.2 | Neuron 2.2.4 |
|---|---|---|---|
| Leukocytes | | | |
| Basophils.Absoluto | | | X |
| RDW | | | |
| Eosinophils.Absoluto | | | |
| Hematocrit | X | | |
| Hemoglobin | X | | |
| Platelets | X | X | X |
| Red.blood.Cells | X | | X |
| Lymphocytes.Absoluto | | X | X |
| MCHC | X | X | |
| MCH | X | | X |
| MCV | | | X |
| Monocytes.Absoluto | | | |

SOM uses vector calculation of object distances to determine to which neuron each object will be related. In this way, the variables have different contributions in each of the neurons, which are formed by the similarities between the groups of objects, always disregarding the labels. Comparing this information with another algorithm that has some *XAI* characteristics, random forest, it is possible to spot the difference. In Table 15, the contribution (weight) of each of the variables in the formation of the random forest model is shown.

**Table 15.** Relevance of variables in the **random forest** training, for comparative purposes.

| Attribute | Relevance % |
|---|---|
| Monocytes.Absoluto | 100.000 |
| Platelets | 99.990 |
| Leukocytes | 91.247 |
| Lymphocytes.Absoluto | 86.264 |
| Red.blood.Cells | 75.683 |
| MCHC | 73.519 |
| Hematocrit | 72.402 |
| MCV | 67.696 |
| MCH | 65.961 |
| Hemoglobin | 58.374 |
| RDW | 58.342 |
| Eosinophils.Absoluto | 51.907 |
| Basophils.Absoluto | 0.000 |

The values presented in Table 15 can be justified because, when assembling the multiple trees within the model, the values of the variables are considered, one variable at a time, and the associated labels, since it is a supervised training algorithm. This causes a bias in the algorithm, and the resulting model ends up ignoring important variables, such as "Eosinophils" and "Basophils." For this reason, it is important to consider the entire object in the analysis, not just individual variables.

## 6. Conclusions and Future Works

With the increasing volume of information generated every day in health centers and hospitals, it is now a demand to use artificial intelligence (AI) techniques to assist in clinical diagnosis and identification of predictive factors of poor or good prognosis, associations with outcomes and comorbidities and death. In laboratory tests, the risk of loss of sensitivity or the generation of a false diagnosis impacting the outcome of the patient has led specialists to become interested in algorithms whose results can also be interpreted by a human. In this sense, AI is a tool for exploring algorithms with the aim of improving

diagnostic recognition rates. Thus, identifying variables that contribute to decision making in clinical diagnoses and allow an understanding of how the algorithm reaches a decision may assist with the search for new treatments and early identification of positive cases until diseases are adequately treated.

In this sense, this paper presented an algorithm that uses the self-organizing map as a base, extending its characteristic of analyzing similarity in data and topological maintenance of characteristics to a hierarchical structure in the form of a tree, which makes it possible to delve into data whose diagnosis is not precise.

The **TESSOM** algorithm allowed analyzing positive cases of COVID-19 from blood test data, serving as an auxiliary tool in identifying variables in the blood count that most impact the algorithmic discrimination between positives and negatives for COVID-19. Thus, in addition to enabling decision support in cases of imprecise diagnosis, it is also possible to identify, for each variable, the range of actual values shown in positive cases, supporting the exploration of visual correlations and aiding in disease monitoring.

To validate the experiments and demonstrate the refinement and robustness of our method, we presented statistical analyses to verify the significance of **TESSOM**'s results.

The inclusion of *XAI* characteristics in **TESSOM** extends results from classic algorithms in the literature, such as **random forest**, which presents only the hierarchy of relevant variables, without relating them in a combination of factors that influence the diagnosis.

As future work, we intended to consolidate the results into a single tool to facilitate user interpretation in terms of usability. Parametric sensitivity analysis, with a theoretical rationale, of **TESSOM**, is also a work to be done in the future, although the results obtained are statistically similar to those of other algorithms. Studies using TESSOM will also be performed using other information, such as socio-economical and cultural distribution, and geographical data, in order to identify variations related to these attributes. The time complexity of the algorithm and the convergence analysis will be analyzed in future work, since the time for convergence is currently a major computational problem, but it was not the focus of this work. Finally, the **TESSOM** applied here is a tool for the inclusion of new indicators and resources to support the decision process and the discovery of new knowledge from the database that can together facilitate interpretation and clinical diagnosis in a diversity of contexts.

**Author Contributions:** Conceptualization, V.S. and L.A.D.S.; data curation, A.A.D.S.; formal analysis, V.S. and A.A.D.S.; investigation, V.S., L.A.D.S. and D.C.D.A.; methodology, V.S. and A.A.D.S.; project administration, V.S.; resources, T.S.B. and R.M.; supervision, L.A.D.S. and A.A.D.S.; validation, L.A.D.S., A.A.D.S., D.C.D.A. and R.M.; visualization, V.S. and A.A.D.S.; writing—original draft, V.S.; writing—review and editing, V.S., L.A.D.S., D.C.D.A., T.S.B. and R.M. All authors have read and agreed to the published version of the manuscript.

**Funding:** This work was partially supported by Fapesp Proc. 2015/24341-7, Brazil, and by Hospital do Coração, Hospital das Clínicas and Hospital Sirio Libanês (São Paulo, Brazil).

**Institutional Review Board Statement:** Not Applicable.

**Informed Consent Statement:** For the purpose of this article, no experiments were performed on humans. All data were collected at the emergency room entrance, with the consent of the patients under the responsibility of the hospitals, and provided anonymously, making it impossible to relate them to a real patient.

**Data Availability Statement:** The data used for this research were made available by FAPESP, in cooperation with the University of São Paulo, with the goal of creating a repository with data related to COVID-19 to enable research on this theme, at https://repositoriodatasharingfapesp.usp digital.usp.br. For this work, the data obtained and treated were made available in the repository https://github.com/vsargiani/DadosCovid2021.git, under MIT license (accessed on 19 march 2022).

**Conflicts of Interest:** The authors declare that they have no conflict of interest.

## Abbreviations

The following abbreviations are used in this manuscript:

| | |
|---|---|
| CBC | Complete Blood Count |
| RDW | Red blood cell Distribution Width |
| MCHC | Mean Corpuscular Hemoglobin Concentration |
| MCH | Mean Corpuscular Hemoglobin |
| MCV | Mean Corpuscular Volume |

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
