# Peer review of "Supporting Clinical COVID-19 Diagnosis with Routine Blood Tests Using Tree-Based Entropy Structured Self-Organizing Maps"

_applsci, doi:10.3390/app12105137_

Round 1
Reviewer 1 Report
1. The abstract is too wordy. The motivation for the study is unclear. Lack of description of the proposed method.
2. The proposed method is simple and lack of innovation.
3. What is the time complexity of the proposed algorithm? Running time is also a big problem.
4. The pseudo code of the algorithm is chaotic.
5. The comparison method is not convincing. The author should add representative methods to the work related to the section and SOM variants (such as btasom).
6. In order to clearly show the process of the proposed algorithm, please add a detailed algorithm flow chart.
7. The author lacks the convergence analysis of the proposed algorithm.
8. There are grammatical errors and tense inconsistencies in English expression.
Author Response
Dear Reviewer,
We thank you for your valuable comments and suggestions. In this document we describe the alterations that were done to the article, according to your comments:
- The abstract is too wordy. The motivation for the study is unclear. Lack of description of the proposed method.
The abstract was reviewed following the Nature guidelines, avaliable at https://cpb-ap-se2.wpmucdn.com/sites.rmit.edu.au/dist/b/55/files/2018/04/Abstract-Guidelines-Nature-Journal-qip46l.pdf
The revised abstract will be as follows:
Data classification is an automatic or semi-automatic process that, utilizing Artificial Intelligence algorithms, learns the variable and class relationships of a data set for use a posteriori in situations where the class result is unknown. For many years, work on this topic has been aimed at increasing the hit rate of algorithms. However, when the problem is restricted to application in health care, besides the concern with performance, it is also necessary to design algorithms whose result is understandable by the specialist responsible for making the decision. Among the problems in the field of medicine, a current research focus is related to COVID-19, in which AI algorithms may contribute to early diagnosis. Among the available COVID data, the blood test is a typical procedure performed when the patient seeks the hospital, and its use in the diagnosis allows reducing the need for other diagnostic tests that can impact the detection time and financial costs. In this work, we propose using Self-Organizing Maps (SOM) to discover attributes in blood test exams that are more relevant for COVID-19 diagnosis. The work applies SOM and entropy calculation in the definition of a hierarchical, semi-supervised and explainable model named TESSOM (Tree-based Entropy Structured Self-Organizing Maps), in which the main feature is enhancing the investigation of groups of cases with a higher level of class overlapping, as far as the diagnostic outcome is concerned. Framing the TESSOM algorithm in the context of Explainable Artificial Intelligence (XAI) makes it possible to explain the results to the expert in a simplified way. It is demonstrated in the work that the use of the TESSOM algorithm to identify attributes of blood tests can help the identification of COVID-19 cases, in addition to having a performance increase of 1.489% in multi scenarios when analyzing a base of 2207 cases from 3 hospitals in the state of São Paulo, Brazil. This work is a starting point for researchers to identify relevant attributes of blood tests for COVID-19 and to support the diagnosis of other diseases.
- The proposed method is simple and lack of innovation.
In order to present the proposed innovation in a clearer way, we included the section Research contributions in the Introduction with the following text:
The main goal of this work is providing a supporting tool for researchers, allowing the analysis of data avoiding bias introduced by the dominant class. In addition to this, the analytical process is performed in a way that is explainable to the specialist. In this way, it is possible to use TESSOM to support the identification of attributes that may contribute to COVID-19 diagnosis, as well as allowing a predictive analysis. Another goal of the model is allowing the comprehension of how the algorithm produced its outcome and enhancing the map resolution through data segmentation. In this way, it is possible to understand the overlapping in the result of blood tests.
Hence, the main contribution of this work is the proposal of a new algorithm that allows the identification of attributes that may contribute to diagnostic explanation, enhancing the comprehension of the obtained results and producing more information that allow the specialist to have more grounding in decision making. Another contribution is the segmentation of the database by similarity, disregarding the object classes. This segmentation makes it possible to identify groups of patients with different diagnostic outcomes, but with similar features.
- What is the time complexity of the proposed algorithm? Running time is also a big problem.
In fact, this is a relevant issue, as the proposal involves a recursive algorithm. However, the aim of this work is presenting the new TESSOM algorithm and the results that can be obtained through its execution. A time complexity analysis may be performed in future works. In order to make this point more clear, the following paragraph was added to the Future works section (in bold):
As future works, it is intended to consolidate the results into a single tool to facilitate user interpretation in terms of usability. Parametric sensitivity analysis, with a theoretical rationale, of TESSOM is also a work to be done in the future, although the results obtained are statistically similar to those of other algorithms. Studies using TESSOM will also be performed using other information, such as socio-economical and cultural distribution, as well as geographical data, in order to identify variations related to these attributes. The time complexity of the algorithm and the convergence analysis will be analyzed in the future work, since time for convergence is currently a major computational problem, but it was not the focus of this work. Finally, the TESSOM applied here presents itself as a tool for the inclusion of new indicators and resources to support the decision process and the discovery of new knowledge from the database that can together facilitate interpretation and clinical diagnosis in a diversity of contexts.
- The pseudo code of the algorithm is chaotic.
To allow a better comprehension of the algorithm, the pseudocode for TESSOM (previously, “Algorithm 3”) was replaced by a flow chart of the process (Figure 3). Also, pseudocode 2, on SOM training, was removed. Text in line 274 was adjusted to reference these adjustments. Algorithm 1 was kept in the text, but its presentation was revised to allow a better understanding.
The flow chart will be presented in item 6 and the text that was replaced in lines 279-284 is the following (in bold):
The algorithm represented by figure \ref{fig:algoTESSOM} works recursively, and whenever the parameters are met by a neuron a new recursive call is made with the objects identified by this neuron as input, This code is responsible for training the model, generates views for the model information, performs some analytical processes, and then performs the model entropy calculation through the algorithm \ref{alg:calcEntropia}. In this way, the model is trained until its conditions are no longer satisfied.
- The comparison method is not convincing. The author should add representative methods to the work related to the section and SOM variants (such as btasom).
The methods chosen for comparison were the same previously applied in COVID data. The BTASOM (“binary tree SOM”) algorithm is used for dynamic growth of the map and was derived from another algorithm named TASOM, that concentrates its analysis on temporal data. The difference between TAMSOM and BTASOM is just the fixed or dynamic number of neurons, and all works found with this algorithms are based on temporal data such as school transcripts or handwriting recognition. Anyway, the justification for our choice was made more evident in line 626, where studies on BTASOM were also included:
The reason for choosing algorithms in Table \ref{tab:classificacaoAlgoritmos} for comparison is the wider use of these algorithms in research on the same topic (analysis of hemogram variables). Other algorithms based on SOM trees are related to this research, but are focused on other types of data, For example, BTASOM is used for analysis of temporal data \cite{Shah-Hosseini2011} or image analysis \cite{Alvarez2015}.
- In order to clearly show the process of the proposed algorithm, please add a detailed algorithm flow chart.
Answered in item 4. The flowchart identified as figure 3 replaced Algorithm 3. The TESSOM process is recursive until one of the two conditions is not satisfied. In this case, it returns a “child” model that is added to the parent node. Then the tree is created in memory in a top-down manner, but its construction is done in a bottom-up manner, adding the inferior levels to the superior ones.
Figure 3. TESSOM Flow Chart
- The author lacks the convergence analysis of the proposed algorithm.
In fact this is an important point that was not yet addressed in the work. This will be added to the future works (see lines 792-803).
- There are grammatical errors and tense inconsistencies in English expression.
The work was revised to remove the inconsistencies.

Reviewer 2 Report
Dear Authors,
Submission entitled:
"Supporting clinical decision for COVID-19 diagnosis from routine blood tests using Tree-based Entropy Structured Self-Organizing Maps "
might fill some knowledge gap in the seminal literature on artificial intelligence applications in infectious diseases.
It is well balanced and pictoresque presenting some truly innovative ideas.
Its core bottleneck weakness is its overtly homogeneous evidence base relying almost exclusively on academic sources from wealthy OECD countries.
Yet, next to California, currently leading competitive nation in AI development is China and its sources alongside with other coming from Emerging BRICS markets inclusive of Brazilian one, should be far more cited.
Expanded and diversified reference track record would give it a sharp edge in terms of transnational comparability and applicability of findings.
For this purpose I warmly recommend consideration for inclusion of the following sources:
https://www.tandfonline.com/doi/full/10.1080/13696998.2021.2013675
https://www.tandfonline.com/doi/full/10.1080/14737167.2020.1823221
https://www.mdpi.com/1660-4601/17/24/9404/htm
I remain willing to review the revised manuscript.
Author Response
Dear Reviewer,
We thank you for your valuable comments and suggestions. In this document we describe the alterations that were done to the article, according to your comments:
Dear Authors,
Submission entitled:
"Supporting clinical decision for COVID-19 diagnosis from routine blood tests using Tree-based Entropy Structured Self-Organizing Maps "
might fill some knowledge gap in the seminal literature on artificial intelligence applications in infectious diseases (...)
Yet, next to California, currently leading competitive nation in AI development is China and its sources alongside with other coming from Emerging BRICS markets inclusive of Brazilian one, should be far more cited.
Expanded and diversified reference track record would give it a sharp edge in terms of transnational comparability and applicability of findings.
For this purpose, I warmly recommend consideration for inclusion of the following sources:
https://www.tandfonline.com/doi/full/10.1080/13696998.2021.2013675
https://www.tandfonline.com/doi/full/10.1080/14737167.2020.1823221
https://www.mdpi.com/1660-4601/17/24/9404/htm
I remain willing to review the revised manuscript.
We thank you for acknowledging the importance of our research. Indeed the comments regarding the infodemic are important and that is why we have included the following paragraph in the introduction (lines 40-43). Finally, we thank you for the suggested references that were used in this new infodemic paragraph.
There are several origins to contribute to the current \textit{Infodemic} related to COVID-19. Some examples come from Asiatic countries that are focused on pharmaceutical innovation \cite{Jakovljevic2021}, studies related to the macroeconomic crisis triggered by the COVID-19 pandemic \cite{Krstic2020} and also studies on the analysis of measures to help combat COVID-19 in the Russian federation \cite{Reshetnikov2020}.

Reviewer 3 Report
This is a well-conceived and organized proposal for conducting predictive modeling of blood-related tests for COVID-19. Four predictive analytical procedures are documented and compared concerning the determination of predictive accuracy. The simulated results are carefully presented.
Overall, it is a very informative paper that could yield useful predictive results for the pandemic disease. One area could be strengthened, however. The predictive power of each analytical approach could be documented and supported by a theoretical rationale, not just by an empirical determination. Furthermore, it would be a good idea to present the sensitivity and specificity of the chosen test procedures for detecting the illness. For instance, a ROC curve could be added.
Author Response
Dear Reviewer,
We thank you for your valuable comments and suggestions. In this document we describe the alterations that were done to the article, according to your comments:
(...)
Overall, it is a very informative paper that could yield useful predictive results for the pandemic disease. One area could be strengthened, however. The predictive power of each analytical approach could be documented and supported by a theoretical rationale, not just by an empirical determination. Furthermore, it would be a good idea to present the sensitivity and specificity of the chosen test procedures for detecting the illness. For instance, a ROC curve could be added.
We thank you for the comments that acknowledge the importance of our research. We agree with the comment on the theoretical rationale of the prediction. However, the focus of the current work is presenting the new algorithm and its practical use. Anyway, your suggestion on developing a theoretical rationale was included as future works (line 792-803).
“As future works, it is intended to consolidate the results into a single tool to facilitate user interpretation in terms of usability. Parametric sensitivity analysis of \textbf{TESSOM}, with a theoretical rationale, is also a work to be done in the future, although the results obtained are statistically similar to those of other algorithms. The time complexity of the algorithm and the convergence analysis will be analyzed in the future work, since time for convergence is currently a major computational problem, but it was not the focus of this work. Finally, the \textbf{TESSOM} applied here presents itself as a tool for the inclusion of new indicators and resources to support the decision process and the discovery of new knowledge from the database that can together facilitate interpretation and clinical diagnosis in a diversity of contexts.”
In addition to this, we included the ROC curve analysis in the Results section, lines 579-582. The following paragraph was added, describing the results obtained with the ROC curves:
With figure \ref{fig:CurvaROCTESSOM} it is possible to evaluate the \textbf{ROC} curve for the TESSOM algorithm, that presents \textbf{AUC} of 0.973 for training (green line) and 0.749 for model tests (prediction - red line). Comparing the obtained curve with those for \gls{MLP}, \gls{knn} and Random Forest, presented in Figure \ref{fig:CurvaROCGeral}, the gain obtained through the use of TESSOM can be identified.
fig:CurvaROCTESSOM = figure 12
fig:CurvaROCGeral = figure 13

Reviewer 4 Report
Dear Authors it is very crucial! It is a future! but , unfortunately at the moment it has crucial limitations. The prior limitations is related to socio-economic and cultural distribution , other limitation is related to different geographical distribution. Please can the authors describe it? Thank you
Author Response
Dear Reviewer,
We thank you for your valuable comments and suggestions. In this document we describe the alterations that were done to the article, according to your comments:
Dear Authors it is very crucial! It is a future! but , unfortunately at the moment it has crucial limitations. The prior limitation is related to socio-economic and cultural distribution, other limitation is related to different geographical distribution. Please can the authors describe it? Thank you
We thank you for the comment. We agree with the suggested analysis. However, the dataset used unfortunately does not provide such information for analysis. We are including the suggestion for future works, in lines 795-797:
Studies using TESSOM will also be performed using other information, such as socio-economical and cultural distribution, as well as geographical data, in order to identify variations related to these attributes.

Reviewer 5 Report
The manuscript under review is a very interesting study aimed to evaluate the potential use of a self-organizing map (called TESSOM) to identify variables related to the diagnosis of COVID-19. The algorithm produces a hierarchical tree. The tree generation and interpretation are well explained and detailed.
I have only a few minor comments:
- In “methods” section, I suggest using the present or the past, but not the future tense.
- In “methods” section, the “list of blood factors that are part of this study” includes some information that the scientific community readers should already know. For example, the definition of each factor (hematocrit is the “calculation of percentage of red blood cells…” ecc) is redundant, the paper has been proposed to a scientific journal, so the readers should have a basic scientific preparation. I suggest synthetizing this list, indicating only how these factors could be useful in COVID-19 patients (e.g., “Hematocrit: its drop may indicate respiratory problems and reflect the severity in cases of COVID-19”).
- The reference list should be implemented with the following papers: 10.1016/j.dsx.2020.04.012; 10.3390/medicina58020144; 10.1016/j.jaip.2020.07.003; 10.3390/vaccines10020308; 10.1016/S2589-7500(20)30295-8
Overall, I think this work is good.
Author Response
Dear Reviewer,
We thank you for your valuable comments and suggestions. In this document we describe the alterations that were done to the article, according to your comments:
The manuscript under review is a very interesting study aimed to evaluate the potential use of a self-organizing map (called TESSOM) to identify variables related to the diagnosis of COVID-19. The algorithm produces a hierarchical tree. The tree generation and interpretation are well explained and detailed.
I have only a few minor comments:
- In “methods” section, I suggest using the present or the past, but not the future tense.
- In “methods” section, the “list of blood factors that are part of this study” includes some information that the scientific community readers should already know. For example, the definition of each factor (hematocrit is the “calculation of percentage of red blood cells…” ecc) is redundant, the paper has been proposed to a scientific journal, so the readers should have a basic scientific preparation. I suggest synthetizing this list, indicating only how these factors could be useful in COVID-19 patients (e.g., “Hematocrit: its drop may indicate respiratory problems and reflect the severity in cases of COVID-19”).
- The reference list should be implemented with the following papers: 10.1016/j.dsx.2020.04.012; 10.3390/medicina58020144; 10.1016/j.jaip.2020.07.003; 10.3390/vaccines10020308; 10.1016/S2589-7500(20)30295-8
Overall, I think this work is good.
We thank you for the reading, comments and suggestions for adjustments that were very helpful in the final revision of the article. We also thank you for the suggested references, that were included in the following parts of the text:
10.1016/j.jaip.2020.07.003 - line 48, after “incomprehensible by a specialist.”, as the third citation.
10.1016/j.dsx.2020.04.012 and 10.1016/S2589-7500(20)30295-8 - Line 133, after the expression “ found in the most diverse contexts”
10.3390/vaccines10020308 - Line 134, last citation after “analysis of clinical data”
10.3390/medicina58020144 - Line 468, after “cytokines, increasing the inflammatory status of the disease \cite{Kahn2021}”
Regarding the suggestion on summarizing the explanation about the blood test factors, we opted to keep the style of the explanation because it is used as a conceptual grounding for the discussion of results.

Round 2
Reviewer 1 Report
The quality of the paper has been improved after modification, and there are two minor problems that need to be noted:
- The image on the right in Figure 15 is abnormal.
- As a new algorithm, the theoretical complexity and actual running time of the algorithm are very basic characteristics, which should not be avoided by the author.
Author Response
The quality of the paper has been improved after modification, and there are two minor problems that need to be noted:
Thank you for the positive comment and we take the opportunity to thank you for the review that helps to improve the quality of the work
- The image on the right in Figure 15 is abnormal.
In fact, there was a problem generating the figure because the file name had some special Portuguese language characters. The adjustment was made, and thanks for the comment.
- As a new algorithm, the theoretical complexity and actual running time of the algorithm are very basic characteristics, which should not be avoided by the author.
We appreciate your comment and recognize the importance of presenting the theoretical computational complexity and runtime. We have included in lines 308-324 of the “TESSOM Algorithm” section information about the theoretical computational complexity of the algorithm. The terminology has been adjusted compared to the references to reflect the complexity of TESSOM.
Also, to address the comment, we calculated the computational time whose excerpt below was inserted into the article at lines 600-603 and also table 7.
The following references on computational complexity were included:
Sani, Habiba Muhammad, Ci Lei, and Daniel Neagu. 2018. “Computational Complexity Analysis of Decision Tree Algorithms.” In Lecture Notes in Computer Science (Including Subseries Lecture Notes in Artificial Intelligence and Lecture Notes in Bioinformatics), 11311 LNAI:191–97. Nature Springer. https://doi.org/10.1007/978-3-030-04191-5_17.
Melka, Josué, and Jean-Jacques Mariage. 2017. “Efficient Implementation of Self-Organizing Map for Sparse Input Data.” In Proceedings of the 9th International Joint Conference on Computational Intelligence, 54–63. SCITEPRESS - Science and Technology Publications. https://doi.org/10.5220/0006499500540063.
Lines 308-324( in the “TESSOM Algorithm” section)
“Due to the way in which \textbf{TESSOM} processes the objects of the training set (SOM training and tree organization), the computational complexity $T$ of the algorithm should be defined as a composition of complexity computations for the SOM in conjunction with its tree structure. For the SOM, as defined by \cite{Melka2017}, complexity can be defined as:
T = O(KMDn) (2)
where $K$ represents the number of training epochs, $M$ is the total number of neurons in the SOM grid, $D$ is the number of object dimensions (number of attributes) and $n$ is the number of objects.
To complement the definition of the tree-based SOM (TESSOM), the definition presented by \cite{Sani2018} identifies the complexity equation for decision trees, used in this work as the basis for calculating the asymptomatic complexity, which can be defined as:
T = O(n) + O(mn \log_2 n) + O(n\log_2n) (3)
where $n$ represents the number of objects, and $m$ the number of attributes.
Assuming that $K$, $M$ and $D$ are constants, then the equation 2 can be simplified using $S = (K * M * D)$. Thus, the theoretical computational complexity of the TESSOM algorithm is defined as:
T = O(Sn) + O(mn \log_2(n)) + O(n\log_2(n)) + O(n) (4)
The equation 4 summarizes the theoretical computational complexity for TESSOM, considering the batch processing used by the SOM training, the entropy calculation, and the tree generation.
"
Lines 600-603, in the Results section (After “through the use of TESSOM can be identified”)
“The execution times of each algorithm in seconds are shown in table 7. It is important to note that for prediction, TESSOM presents higher times because it involves two methods, and one of the methods internally has kNN training that uses the objects in the neuron to perform the prediction.”

Reviewer 4 Report
It is ok for me.
Thank you
Author Response
Thank you for taking the time to review our article.